# Quantifying post-transcriptional regulation in the development of *Drosophila melanogaster*

Kolja Becker[1], Alina Bluhm[1], Nuria Casas-Vila[1], Nadja Dinges[1], Mario Dejung[1], Sergi Sayols[1], Clemens Kreutz[2], Jean-Yves Roignant[1], Falk Butter[1] & Stefan Legewie[1]

Even though proteins are produced from mRNA, the correlation between mRNA levels and protein abundances is moderate in most studies, occasionally attributed to complex post-transcriptional regulation. To address this, we generate a paired transcriptome/proteome time course dataset with 14 time points during *Drosophila* embryogenesis. Despite a limited mRNA-protein correlation ($\rho = 0.54$), mathematical models describing protein translation and degradation explain 84% of protein time-courses based on the measured mRNA dynamics without assuming complex post transcriptional regulation, and allow for classification of most proteins into four distinct regulatory scenarios. By performing an in-depth characterization of the putatively post-transcriptionally regulated genes, we postulate that the RNA-binding protein Hrb98DE is involved in post-transcriptional control of sugar metabolism in early embryogenesis and partially validate this hypothesis using Hrb98DE knockdown. In summary, we present a systems biology framework for the identification of post-transcriptional gene regulation from large-scale, time-resolved transcriptome and proteome data.

[1] Institute of Molecular Biology (IMB), Ackermannweg 4, 55128 Mainz, Germany. [2] Center for Biosystems Analysis (ZBSA), University of Freiburg, Habsburger Str. 49, 79104 Freiburg, Germany. These authors contributed equally: Alina Bluhm, Nuria Casas-Vila, Nadja Dinges. Correspondence and requests for materials should be addressed to F.B. (email: f.butter@imb.de) or to S.L. (email: s.legewie@imb.de)

According to the central dogma of molecular biology, protein is translated from mRNA, suggesting that mRNA levels can be predictive of protein concentrations. However, the relationship between mRNA and the concentration of its protein does not follow the simple monotonic correlation that higher mRNA levels always relate to concordantly more protein. This nontrivial mRNA−protein relation is a general phenomenon ranging from yeast to human (reviewed for example in refs. [1–3]), that could arise from extensive post-transcriptional gene regulation in eukaryotic organisms. One important mechanism of post-transcriptional gene regulation is controlled protein translation. In line with widespread translational regulation, global studies in mammalian cells showed very different translation rates across mRNAs[4]. Accordingly, ribosome occupancy of a transcript is a better predictor of protein expression when compared to its mRNA concentration[5]. Furthermore, proteins are subject to active degradation via the Ubiquitin-proteasome system, controlling protein degradation rates independently of its transcript abundance[6,7].

Even in the absence of regulated protein translation or turnover, a mismatch between mRNA and protein levels can occur, due to the temporal delay that occurs when protein is translated from mRNA. Accordingly, it has been noted that while mRNA−protein correlations are generally low, the discrepancy is even more pronounced during dynamical cellular transitions[3]. Theoretical considerations revealed that delayed protein dynamics relative to mRNA results in a nonlinear relationship between the two species[8–10], and is especially pronounced for stable proteins with long half-lives[11]. Furthermore, during cellular transitions active regulation of protein translation and turnover is enforced. For instance, RNA binding proteins, micro-RNAs, or RNA modifications can affect processing, translation, and/or turnover of a transcript[12–16].

*Drosophila* embryogenesis poses an interesting case to study post-transcriptional regulation, since almost no transcription takes place during the first 3 h after oocyte fertilization. Instead, developmental dynamics rely on post-transcriptional regulation of maternally deposited mRNA and protein until the transcription of embryonic genes is switched on during maternal-to-zygotic transition (MZT)[17,18]. For example, it is well established that the maternally deposited positional information genes *hunchback*, *caudal*, *nanos*, and *oskar* are regulated post-transcriptionally[19,20]. Global measurements of ribosome occupancy at various time points demonstrated that post-transcriptional regulation is a widespread phenomenon in the developing embryo[21]. Given that RNA-binding proteins (RBPs) are major regulators of post-transcriptional gene regulation, it is not surprising that the mRNA-bound proteome is highly dynamic during MZT[22].

These complex dynamics of post-transcriptional regulation make it difficult to intuitively understand the relations between mRNA and protein expression changes. Mechanistic models of protein translation provide a possible mathematical framework for an improved understanding of gene expression regulation. For instance, using simple models of protein translation describing the spatio-temporal mRNA−protein relationship of three *Drosophila* gap genes, we concluded that protein abundance merely represents a time-delayed version of its corresponding mRNA not requiring post-transcriptional regulation[8]. On a genome-wide level, Teo et al. and Cheng et al. developed a time-discrete model of protein turnover and directly inferred post-transcriptional regulation from characteristic features of protein and mRNA time courses[23,24]. Applying kinetic, time-continuous, models of protein translation, Peshkin et al. concluded for *Xenopus laevis* development that the majority of protein expression changes can be explained by assuming proportional protein production from

mRNA and first-order protein degradation[25]. Likewise, in dendritic cells responding to external lipopolysaccharide treatment, Jovanovic et al. conclude that simple models of protein translation explain most of the variance in protein expression based on mRNA abundance changes[26].

In this work, we investigate the relation between mRNA and protein levels during *Drosophila* embryogenesis using highly time-resolved paired transcriptome/proteome measurements. Compared to the previous modeling studies, we specifically describe a framework for the systematic discovery of post-transcriptional regulation mechanisms controlling a biological process of interest: We combine model fitting and model rejection to explicitly name mRNA−protein pairs insufficiently described by four distinct model variants, thereby deriving lists of potentially post-transcriptionally regulated proteins. We show that post-transcriptionally regulated proteins are enriched for certain biological functions, including cell cycle and regulation of glucose metabolism. Furthermore, based on sequence motif analyses and knockdown experiments, we propose that the splicing factor Hrb98DE may be involved in post-transcriptional control of glucose metabolism.

## Results

**Paired mRNA/protein data reduces experimental variation**. We previously followed proteome changes during *Drosophila* development with high temporal resolution[27]. This dataset included measurement time points every 1 h within the first 6 h after egg-laying, and every 2 h hereafter until 20 h (Fig. 1a). When comparing our proteome dataset to a published developmental RNA-Seq time course[28], we observed limited correlation between RNA and protein levels. To further investigate post-translational gene regulation in more detail, and to exclude that the discrepancies between RNA and protein levels arose from experimental variation between laboratories, *Drosophila* strains, and measured time points, we generated a new RNA-Seq dataset using the same fly embryo samples as for our proteome measurement (Fig. 1a and Supplementary Figure 1a).

Our transcriptome matches the published RNA-Seq dataset at similar stages of development when calculating pairwise correlations of all common reads between samples (Supplementary Figure 1b). Time-course analysis however showed altered developmental speed between our data and the published dataset (Supplementary Note 1 and Supplementary Figures 1b-1d), indicating that laboratory conditions can affect the overall kinetics of gene expression even for a robust biological process as *Drosophila* embryonic development. Thus, pairing of RNA and proteome measurements from the same laboratory avoids the risk of systematic mis-estimations.

**Proteome and transcriptome changes show limited correlations**. We related transcriptome and proteome measurements to explore whether both gene expression layers exhibit a high degree of coordination during *Drosophila* embryogenesis. We based our analysis on median values across four replicates and focused on 3761 RNA−protein pairs detected reproducibly in at least ten time points (Supplementary Data 1).

We classified the temporal behavior of these 3761 mRNA−protein pairs into four groups with qualitatively distinct dynamics using a hierarchical clustering approach (see Fig. 1b). These four groups are: (1) mRNA and protein levels increase concordantly (525 genes), (2) mRNA and protein levels decrease concordantly (1063 genes), (3) mRNA increases, while protein levels decrease (372 genes) and (4) mRNA decreases, while protein levels increase (1801 genes). In total, 58% of genes showed inverse abundance changes between mRNA and protein,

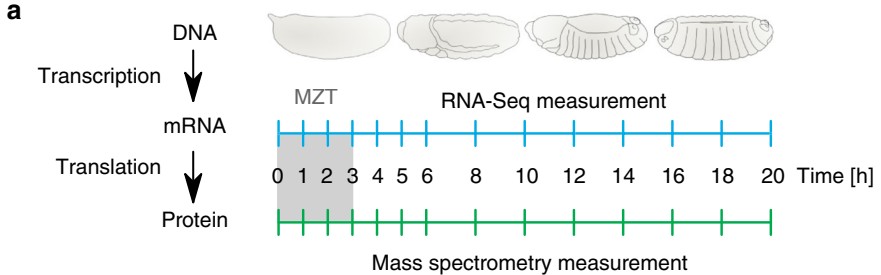

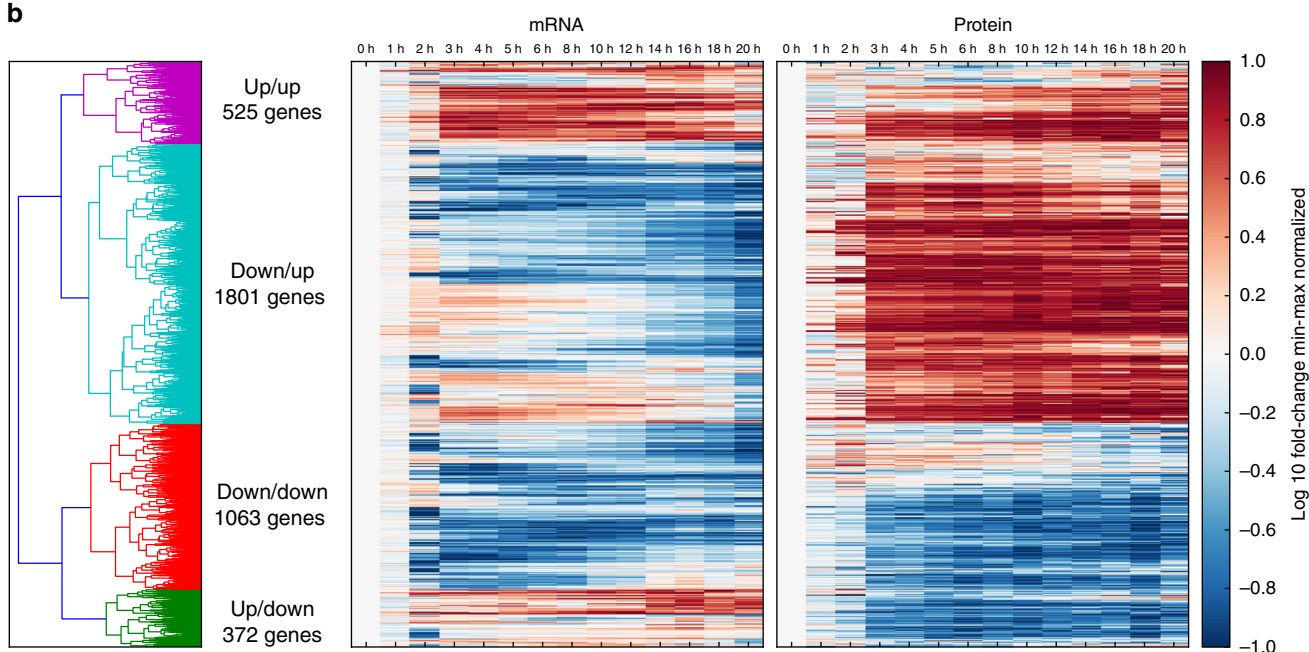

**Fig. 1** Paired transcriptome and proteome of *Drosophila* embryogenesis. **a** Time-points of paired mRNA and protein measurements during *Drosophila* embryonic development using RNA-Seq and mass spectrometry, respectively. The initial time point (0 h) represents egg deposition; the maternal-to-zygotic transition (MZT) occurs within the first 3 h of development. **b** Heatmaps of mRNA and protein time courses. The 3761 mRNA/protein pairs (*y*-axis), for which protein could be quantified in at least 10 of 14 time points, are shown for the developmental time points (*x*-axis). The color code indicates the mRNA and protein fold-changes relative to *t* = 0 h after min-max normalization between −1 and 1. Time courses are sorted according to hierarchical clustering using the Euclidean distance (see also dendrogram on the left). Time courses within each of the four clusters (green, red, blue, purple) roughly follow similar dynamics, reflecting concordant or opposing mRNA and protein dynamics (increase (up) or decrease (down))

indicating that the temporal expression dynamics of most mRNA transcripts and the corresponding proteins are not directly related.

To quantify the relationship between mRNA and protein abundance, we calculated Spearman correlation coefficients globally, relating mRNA and protein levels across all genes at the same time point. We included only proteins showing the largest fold-changes compared to 0 h (*n* = 500) to avoid that random variations mask existing trends between mRNA and protein. In line with previous studies, the average correlation between mRNA and protein levels of the same time point is limited ($\rho = 0.41$) (Fig. 2a, diagonal) with a maximum at 14 h ($\rho = 0.57$) (Fig. 2b—green panel). Of note, this limited correlation is not due to experimental variation, as the mean correlation across four biological replicates at 14 h is $\rho = 0.96$ for the transcriptome and $\rho = 0.96$ for the proteome.

To account for time delays associated with mRNA processing and translation, we also assessed nonsynchronous correlations by globally relating mRNA and protein levels at different time points. In this scenario, the highest mRNA−protein correlation reaches $\rho = 0.63$ between mRNA levels at 12 h and protein abundances at 16 h (Fig. 2b—orange panel). Despite this modest

value, we observed a general trend of better transcriptome/proteome correlation when relating mRNA samples to later protein time points, most likely due to delays in protein synthesis. Furthermore, higher correlations are generally observed at later developmental time points, possibly because less pronounced dynamical changes of mRNA and protein occur after MZT.

To further test for the correspondence of mRNA and protein dynamics, we calculated the correlation between the mRNA and protein time courses for each gene individually. Only a subset of mRNA−protein pairs (*n* = 429 (11.4%)) exhibits a significant positive Spearman correlation coefficient (Fig. 2c—bottom panel, Student's *t* test, two-sided, $p < 0.05$), and the median correlation coefficient over all genes is close to zero (Fig. 2c—top panel). To account for protein expression delays, we additionally calculated the same correlation coefficients after introducing a time shift between mRNA and protein measurements. These shifts only marginally improved the median correlation coefficient over all genes, with a maximum median correlation when the protein was assumed to lag behind the mRNA for 4−6 h (Fig. 2c—top panel). At a time shift of 4 h, we also observed the highest number (*n* = 520 (13.8%)) of mRNA−protein pairs with significant positive correlation (Fig. 2c—lower panel, Student's *t* test, two-

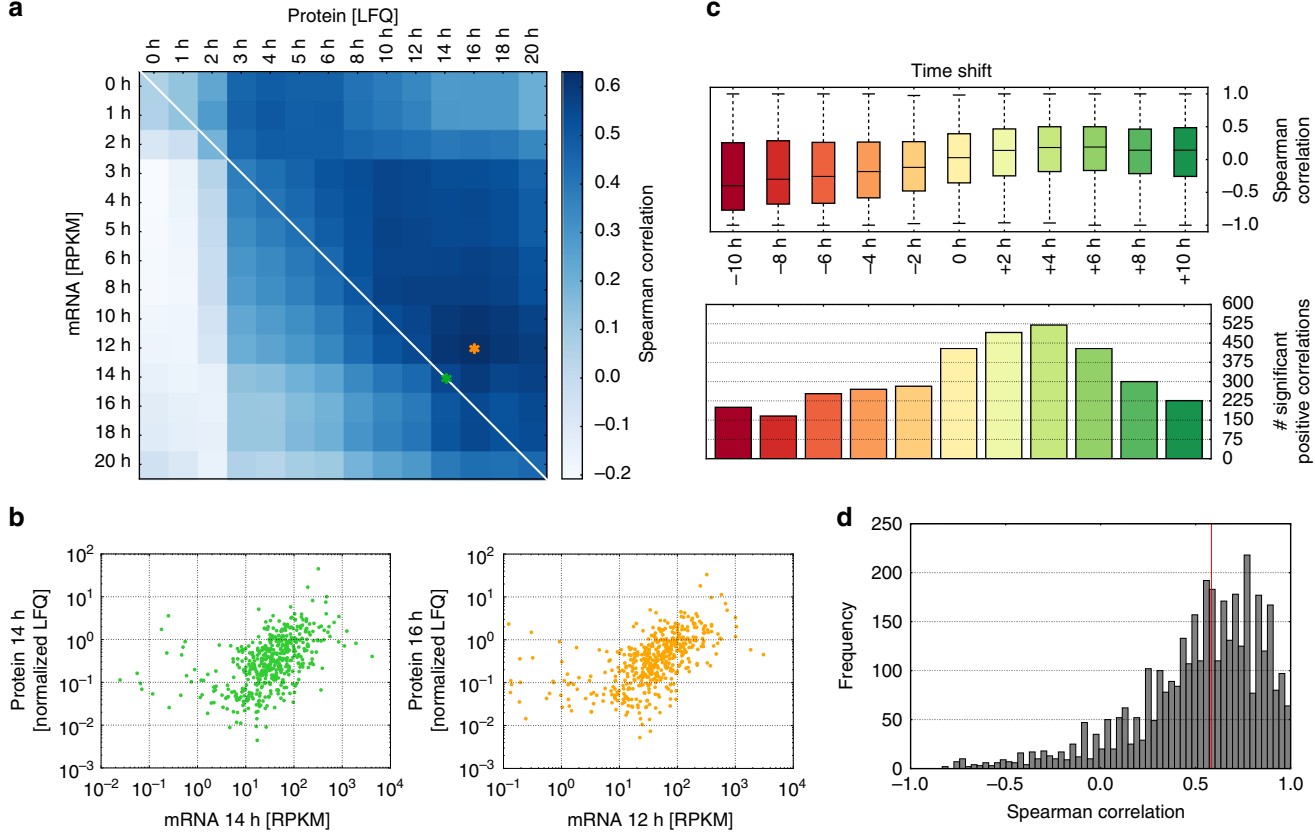

**Fig. 2** Limited correlation between mRNA and protein levels. **a** Global RNA−protein correlation across all samples. Heatmap of Spearman correlation coefficients between protein abundance (*x*-axis; quantified using MaxLFQ), and mRNA levels (*y*-axis; expressed as RPKM values), at the same (diagonal) or shifted (off-diagonal) time points of embryonic development. Only the top 500 proteins showing largest absolute fold-changes compared to the 0 h time point were considered. Green and orange stars indicate maximum correlation between same and shifted time points, respectively. Corresponding scatter plots are shown in (**b**). **b** Maximum global RNA−protein correlation. Scatter plots showing correlation of mRNA (RPKM) and protein (LFQ) levels at 14 h (left, $\rho = 0.56$), or between mRNA at 12 h and protein at 16 h (right, $\rho = 0.63$). Chosen time points correspond to the orange and green stars marked in (**a**). **c** Local correlations relating the mRNA and protein time courses of individual genes were calculated using the Spearman correlation coefficient. Boxplots indicate the distribution of Spearman correlation coefficients for all 3761 mRNA−protein pairs (black line: median; boxes: quartiles; whiskers: 95-percentile) (upper panel). Time shifts between mRNA and protein dynamics were introduced by adding a constant to the time axis of the protein (0 h: no shift). Positive and negative values reflect that protein lags behind or is advanced relative to its mRNA, respectively. Number of mRNA−protein pairs with a significant (Student's *t* test, two-sided, $p < 0.05$), positive correlation for each time shift (bottom panel). **d** Distribution of maximum Spearman correlation coefficients across all time shifts. For each individual mRNA−protein pair ($n = 3761$), the maximum correlation between the mRNA and protein time courses at any time shift between 0 h and +10 h was determined and considered in this histogram. The red line indicates the median of all correlation coefficients

sided, $p < 0.05$). A similar time delay of 2−6 h between peaking of circadian mRNA and protein was also reported by Robles et al.[29].

Taken together, these data indicate a poor correlation of mRNA and protein dynamics. Overall, only 33.7% ($n = 1268$) of genes show a significant (Student's *t* test, two-sided, $p < 0.05$) positive monotonic relationship between mRNA and protein at positive time shifts with protein lagging behind mRNA. As each mRNA−protein pair may be characterized by a distinct delay, we further selected the maximal correlation estimate of each mRNA−protein pair over all time shifts, regardless of significance. This procedure improves the median correlation across all genes to $\rho = 0.58$ (Fig. 2d).

For the analysis of proteomics data, different quantification procedures have been proposed: "intensity-based absolute quantification" (iBAQ[4]) and "label-free quantification" (e.g. MaxLFQ[30]). We found that the conclusion of a limited correlation between mRNA and protein remains valid irrespective of iBAQ or MaxLFQ quantification (see Supplementary Note 2 and Supplementary Figure 2). Taken together, this suggests that the relation between mRNA and protein abundances requires a more elaborate mathematical framework than correlation analysis, incorporating gene-specific parameters. We therefore turned to mathematical modeling to better describe the mechanisms underlying protein production from mRNA.

**Kinetic models quantitatively relate mRNA/protein dynamics.** We used a set of dynamical models describing protein expression based on ordinary differential equations (ODEs) and investigated whether the dynamics of each protein can be explained as a function of the corresponding mRNA time course. By fitting these models to the experimentally measured protein expression time courses, we assigned each mRNA−protein pair to a model representing one of four regulatory scenarios described below (schematically depicted in Fig. 3a), or excluded it from any of these.

The default model (production) follows the simple assumption that protein is synthesized from mRNA and additionally subject to degradation. The mRNA concentration serves as an input and proportionally affects the translation rate, implying the absence of

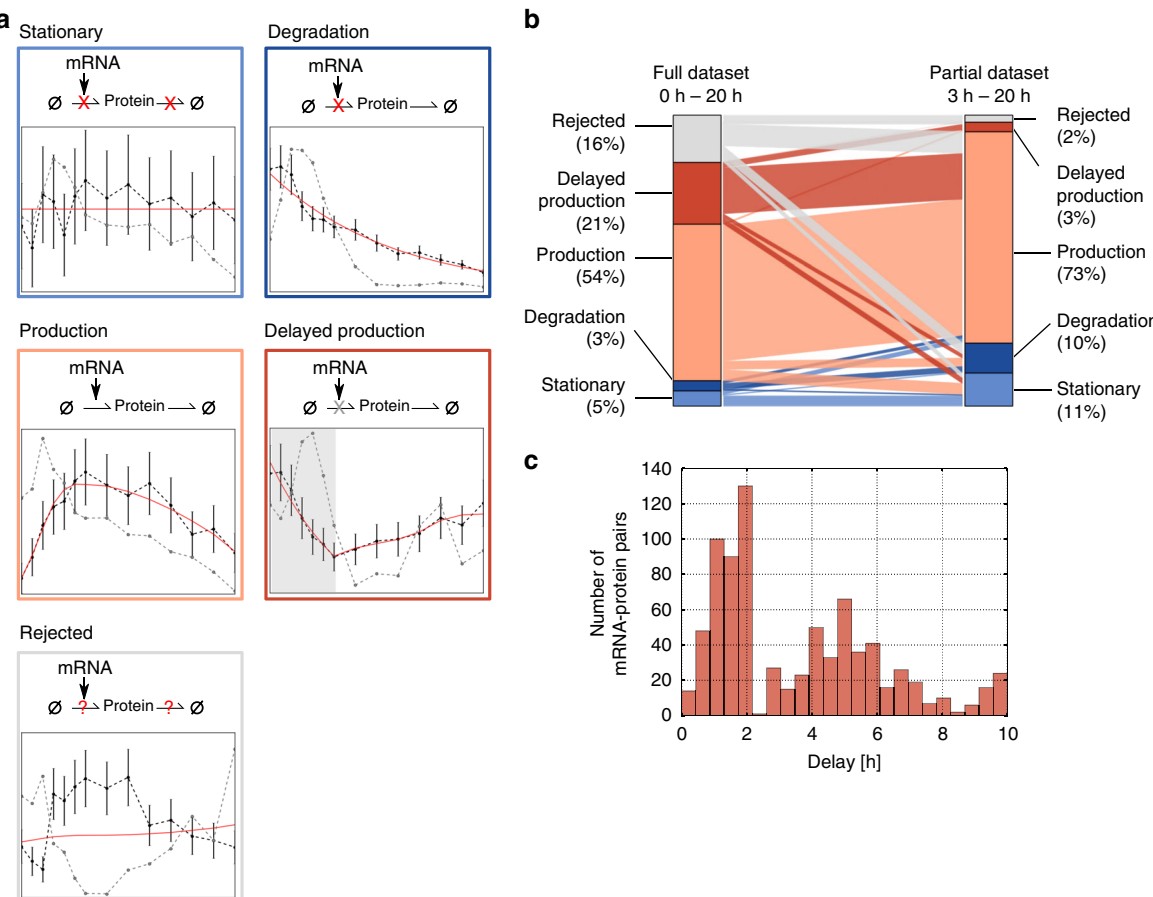

**Fig. 3** Kinetic models quantitatively relate mRNA and protein dynamics. **a** Schematic representation of model variants incorporating protein synthesis and degradation (thin arrows). Red and gray crosses indicate absence or delayed onset of individual reaction steps, respectively. The measured mRNA time courses were used as a model input, and simulated protein output was fitted to the corresponding experimental data by tuning the kinetic parameters. Each of the four different model variants was fitted separately and the best model was selected (see Methods). If all four models were rejected, the protein was classified as potentially post-transcriptionally regulated. An exemplary model fit for each class is given (red line), alongside with corresponding mRNA (gray) and protein (black) expression. Error bars represent standard deviation in protein according to the chosen linear error model. **b** Model-based classification results for 3761 mRNA−protein pairs. Barplot showing fractions of proteins in each class for the full dataset (left; 0−20 h) or post-MZT only (right; 3−20 h). Connecting lines indicate the migration of one protein from one model class to another between both scenarios. **c** Distribution of estimated delay times of 800 proteins assigned to the delay model in which protein translation occurs only with a lag time after egg deposition

complex post-transcriptional regulation. We also considered a delayed-production model, which assumes a temporal delay in translation. In this model, the protein is initially degraded until translation sets in, from which point the delayed-production model corresponds to the production model. The implementation of the delayed-production model was motivated by very low protein accumulation prior to the MZT (see Fig. 1b). Additionally, we assumed two mathematically less complex models: In the degradation model de novo translation can be neglected throughout embryonic development and therefore protein synthesis was omitted. In the stationary model, the protein is assumed to be stable during embryogenesis thereby eliminating protein production and degradation altogether.

The unknown model parameters such as protein synthesis and degradation rates, initial protein concentrations, and delay times were determined by fitting our models to the experimental data. We fitted each of the 3761 mRNA−protein pairs individually using the four model variants, and classified the genes into one of the regulatory scenarios (stationary, degradation, production and delayed-production) using a two-step strategy: First, we assessed whether a given model describes the measured mRNA−protein dynamics based on the difference between model fit and data

using a $\chi^2$-test (Benjamini−Hochberg (BH) corrected $p$ value < 0.05). To exclude that deviations between model and data are correlated in time, we further applied a Durbin−Watson test (BH corrected $p$ value < 0.05), and only considered a model variant feasible if both tests were passed. Second, for mRNA/protein pairs fitting multiple models, we performed a stepwise likelihood-ratio test, balancing the goodness-of-fit against the risk of overfitting, to select for the simplest model explaining the data. If the model selection assigned a protein to the degradation or stationary classes, but the mathematically more complex production model could not be rejected with a nonzero protein translation rate, we re-assigned the protein to the production class. Thus, we consider the production model as the simplest, unregulated protein expression scenario in biological terms, whereas the competing (mathematically simpler) models require the existence of an additional biological factor blocking translation or degradation. While Fig. 3a shows examples of classifications, see Supplementary Note 3 for a detailed description of the classification procedure.

Slightly more than half (54%) of mRNA−protein expression patterns follow the production model, which assumes continuous protein synthesis from mRNA, and are thus likely determined by

pure transcriptional control (Fig. 3b). Furthermore, 5% of proteins were stable during embryogenesis (stationary model), and another 3% of proteins were best fitted by the degradation model. Finally, the assumption of a delay time for translation (delayed-production model) explained another 21% of genes. Altogether, 84% of genes were explained by any of these four simple regulatory models, suggesting that simple ODE models quantitatively describe proteome dynamics by including transcriptome data. For each of the four models the Pearson correlation between fitted and measured protein expression values across all genes and time points is above 0.99 (Supplementary Figure 3).

For the remaining 16% of protein−mRNA pairs all four proposed models were rejected, suggesting that these proteins are under complex post-transcriptional control. Accordingly, we find a statistically significant overlap (1.85-fold enrichment, hypergeometric test, $p = 2.3\mathrm{e}{-16}$, Supplementary Figure 4a) with a published set of translationally regulated developmental genes identified by ribosome profiling[21]. Potentially post-transcriptionally regulated proteins are found in all four dynamical groups obtained by hierarchical time-course clustering (Supplementary Figure 4b). This demonstrates the complementarity of our modeling analysis and suggests that post-transcriptional regulation occurs by various mechanisms rather than being controlled by a global factor. Interestingly, previous proteomic studies report similar fractions of 20−30% of potentially post-transcriptionally regulated proteins for other biological processes (see Discussion). The fact that we obtain slightly smaller numbers of post-transcriptionally regulated genes (16%) most likely reflects that our approach produces a conservative estimate as we show using in silico benchmarking (see Supplementary Note 1).

During the first 3 h until MZT, no transcription occurs in the fly embryo[18]. Thus, any protein must be maternally deposited or translated from maternally deposited RNA. MZT is visible in our data as striking changes in mRNA as well as protein expression patterns at 3 h (Fig. 1b). Since MZT marks the advent of transcriptional activity, we expected a large fraction of post-transcriptional regulation specifically within the first 3 h of development. To test this hypothesis, we repeated the protein classification including only data from time points beyond MZT (3−20 h). In line with a strong decline of post-transcriptional control after MZT, we found that for 98% of proteins the post-MZT data can be explained by one of the four ODE model variants (Fig. 3b, right).

Among these 98%, only 3% of proteins (initially 21%) are still assigned to the delayed-production model, i.e. only few proteins show delayed translation post-MZT. This observation again supports that MZT is a major point of post-transcriptional control at which mRNA translation may be activated on demand[31]. In line with this hypothesis, estimated delay times in the delayed-production model fitted to the full dataset (0−20 h) show a bimodal distribution with a group of proteins exhibiting delays of 0.5−2.5 h (381 proteins) and another group with translation delayed even longer for about 3–10 h (452 proteins) (Fig. 3c). This indicates two waves of translation: a first translation burst coinciding with MZT and a secondary delayed translation phase. The general ability of our classification method to distinguish between different mRNA/protein dynamics is shown in Supplementary Figure 4c.

**Kinetic models account for lack of mRNA−protein correlation**. Given that 84% of mRNA−protein pairs are described without assuming complex post-transcriptional regulation, the low mRNA−protein correlation might be surprising. The proposed

mathematical models provide mechanistic insights into the missing correlation: For the stationary, degradation, and delayed-production classes, the model predicts simple biological control mechanisms on protein translation, leading to an uncoupling of mRNA−protein dynamics.

However, even for the production model, where such control mechanisms are absent, mRNA and protein time courses are not necessarily correlated, as evidenced by a broad distribution of the corresponding correlation coefficients (Fig. 4a, top panel). The lacking correlation seems to arise, in part, because proteins with long half-life lag behind corresponding mRNA[32], hereby decreasing mRNA−protein correlation[33]. Based on our fitting results, we obtained estimates for the kinetic parameters and indeed find that the lack of positive mRNA−protein correlation ($\rho = 0.03$) is particularly evident when considering only stable proteins whose half-life confidence interval exceeds the median (8.4 h) of all protein half-lives (260 proteins; $\rho = -0.55$, Fig. 4a, second panel). In contrast, positive correlations are much more common in the inversely defined group of proteins with short half-lives (331 proteins, $\rho = 0.24$, Fig. 4a, third panel), supporting that protein degradation rates determine how closely protein dynamics follow the mRNA time course.

However, our analysis indicates that even proteins with short half-lives can show a weak mRNA−protein correlation (Fig. 4a, third panel, short half-life). Our model explains many of these mRNA−protein discrepancies with out-of-steady-state protein concentrations at the initial time-point ($t = 0$ h): Certain mRNAs deposited in the egg remain untranslated in the inactive egg and are only translated upon fertilization[34]. Here, initial protein production at $t = 0$ exceeds protein degradation, leading to a net increase in protein level even at constant mRNA amounts. In an extreme case, this can result in an inverse protein−mRNA relationship during the developmental time course, where the protein level increases, while mRNA abundance decreases (Fig. 4b, left). Likewise, mRNA increase with a concomitant protein decay can occur if net protein degradation exceeds synthesis (Fig. 4b, right).

To test whether a large fraction of proteins is out of steady-state at egg deposition ($t = 0$ h), we compared the actual protein expression values with the theoretical steady-state predicted by the ratio of the fitted synthesis and degradation rates (see Fig. 4c). As expected, the correlation of measured protein levels with the theoretical steady-state is limited at fertilization, whereas they agree better later in development. Such a perturbation from steady-state indeed weakens the mRNA−protein correlation: If we consider only proteins with fast dynamics (short half-life), whose abundance is close to their estimated steady-state at 0 h (101 proteins), the median correlation between individual mRNA−protein pairs significantly increases to $\rho = 0.45$ (Fig. 4a, bottom panel) compared to $\rho = 0.24$ for all fast changing proteins (two-sided Kolmogorov−Smirnov test, $p = 0.003$). In a dynamical context, a prerequisite for a high correlation between mRNA and protein is therefore not only fast protein turnover, but also an appropriate initial mRNA−protein ratio with balanced production and degradation rates.

**Regulatory classes show enrichment for biological functions**. We performed Gene Ontology (GO) analyses to investigate whether certain biological functions are specifically represented in each class (Fig. 5). Proteins assigned to the production model showed enrichment for processes related to mRNA processing, splicing and RNA metabolism, suggesting that this class contains critical regulators of post-transcriptional gene expression. Moreover, proteins in the delay group are enriched for GO terms related to protein catabolic processes, possibly indicating

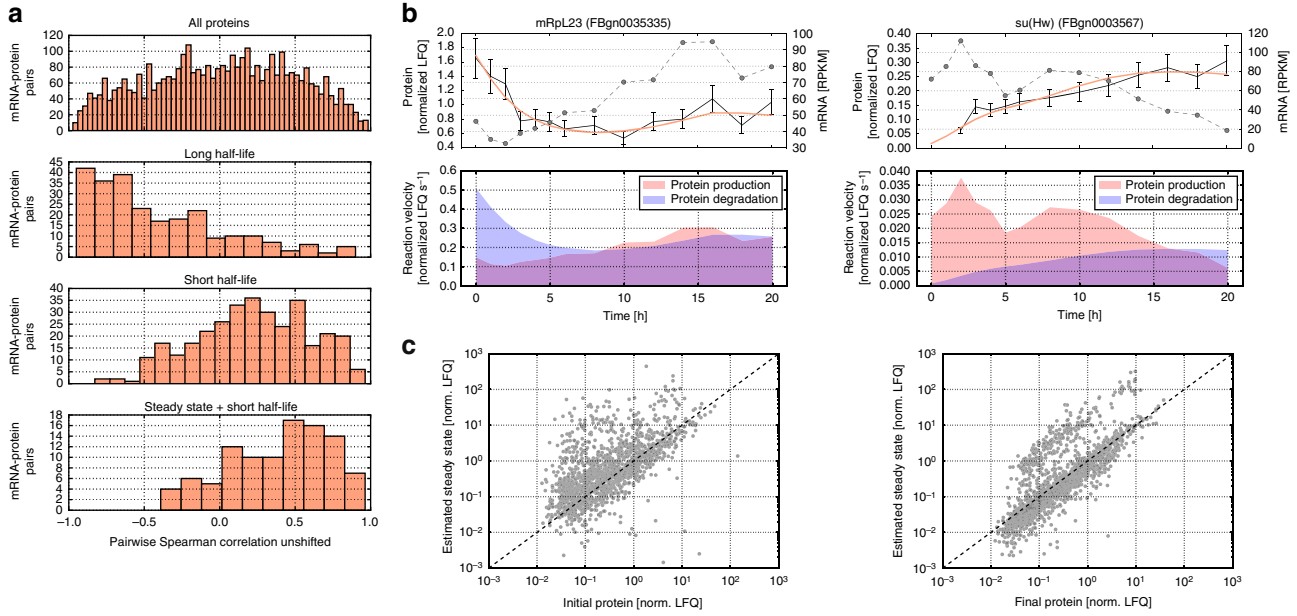

**Fig. 4** Kinetic models explain lack of mRNA−protein correlation. **a** The correlation of mRNA and protein time courses depends on protein dynamics and initial expression levels. Histograms for the distribution of Spearman correlations for all 3761 mRNA−protein pairs (top panel, $n = 3761$), proteins with long half-life (second panel, $n = 260$), proteins with short half-life (third panel, $n = 331$) and proteins with short half-life close to their estimated steady-state at the onset of embryogenesis (bottom panel, $n = 101$). See main text for details. **b** Examples of genes with inverse mRNA and protein dynamics. Left: For mRpL23 (FBgn0035335), mRNA level (dashed line) increased while protein levels (black line) decreased. Error bars represent standard deviation of protein according to the chosen linear error model. Model fit is shown as colored line. The lower panel shows the simulated velocities of protein production and degradation, which are initially unbalanced. Right: Conversely, for su(Hw) (FBgn0003567) the protein production velocity initially exceeds that of degradation, protein amount (black line) increased, while mRNA level decreases. **c** Protein levels early in development tend to deviate from the model-predicted protein steady-state. Measured protein levels for mRNA−proteins pairs classified into the production model ($n = 2027$) are plotted against estimated protein steady-states. Steady-state estimates were derived from the fitted model parameters as well as given mRNA concentrations at 0 h (left panel, Pearson correlation $\rho = 0.61$) and 20 h (right panel, Pearson correlation $\rho = 0.79$) using the equation $\alpha/\lambda \ast \text{mRNA}[t]$ ($\alpha$: production rate; $\lambda$ degradation rate and $t$ time)

widespread expression of factors mediating the degradation of maternal protein at the onset of MZT[35]. Among post-transcriptionally regulated proteins for which all of our four models needed to be rejected we found an enrichment of cell cycle-related genes. Indeed, early nuclear divisions in *Drosophila* embryos cannot be controlled transcriptionally, since transcription is virtually absent before MZT. Accordingly, widespread post-transcriptional regulation has been reported for maternal mRNAs involved in cell cycle regulation[36]. In addition, GO-terms related to sugar metabolism are enriched in the group of potentially post-transcriptionally regulated proteins, which agrees with previous evidence showing post-transcriptional control of genes functioning in glucose metabolism[29].

The mode and time scale of post-transcriptional regulation, however, appears to be distinct for cell cycle and sugar metabolism: Protein levels of genes related to sugar metabolism predominantly upregulate 3 h after egg-laying and afterwards remain nearly stable, while their mRNAs sharply decrease already within 2 h after egg-laying (Supplementary Figure 5a). This suggests early post-transcriptional regulation of sugar metabolic processes before or at MZT. In contrast, a subset of proteins related to the cell cycle shows an abrupt downregulation of both mRNA as well as protein at 14 h, potentially indicating (post-) transcriptional regulation long after the completion of MZT (Supplementary Figure 5b).

**Identifying mechanisms of post-transcriptional regulation**. To uncover potential regulators of post-transcriptional gene

regulation, we searched for enriched sequence motifs in the mRNA sequence of proteins in our protein classes. We considered 67 *Drosophila*-specific RNA -binding protein (RBP) motifs corresponding to 51 different RBPs[37]. We found no or little enrichment of RBP motifs in stationary, degradation, and delayed-production classes, whereas seven RBPs showed significant motif enrichment for the class of potentially post-transcriptionally regulated proteins (Fig. 6a).

Among the enriched motifs were the RBPs Pumilio and Bruno (also known as Aret), which are known post-transcriptional regulators of positional information genes in early *Drosophila* development[19,20]. The strongest motif enrichment was observed for Hrb87F and Hrb98DE (also known as Hrp36 and Hrp38), two proteins recognizing highly similar RNA sequence motifs (Fig. 6a). In our time course, Hrb87F and Hrb98DE both show a ~5-fold upregulation in protein expression (Fig. 6b), indicating that they are developmentally regulated. Accordingly, a recent RNA interactome study demonstrated that the mRNA-bound fraction of Hrb87F and Hrb98DE changes during MZT in *Drosophila*[22]. Interestingly, the known functions of Hrb87F and Hrb98DE match with results of our GO term analysis, as these proteins have been identified in large-scale screens for regulators of the cell cycle and sugar metabolism, respectively[38,39].

**Hrb98DE-dependent regulation of glucose metabolism**. For further analysis, we focused on Hrb98DE, because this protein (and its vertebrate homolog hnRNPA1) has been implicated in the regulation of protein translation[40]. To assess the role of

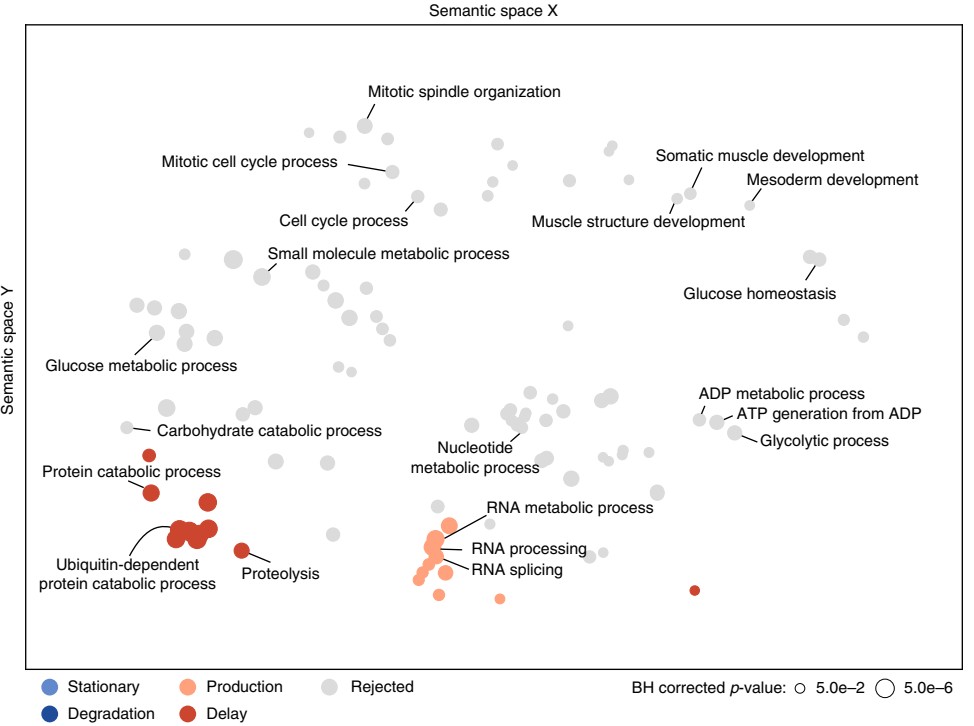

**Fig. 5** Protein classes are enriched for specific biological functions. Distinct GO terms are significantly (hypergeometric test, BH corrected $p$ value < 0.05) overrepresented in the production and delay category as well as for the rejected class (indicated by color). GO terms were arranged according to their semantic similarity and a 2D projection was generated via multidimensional scaling. Each circle represents a single GO term and the size of each circle is proportional to the corrected $p$ value (see legend)

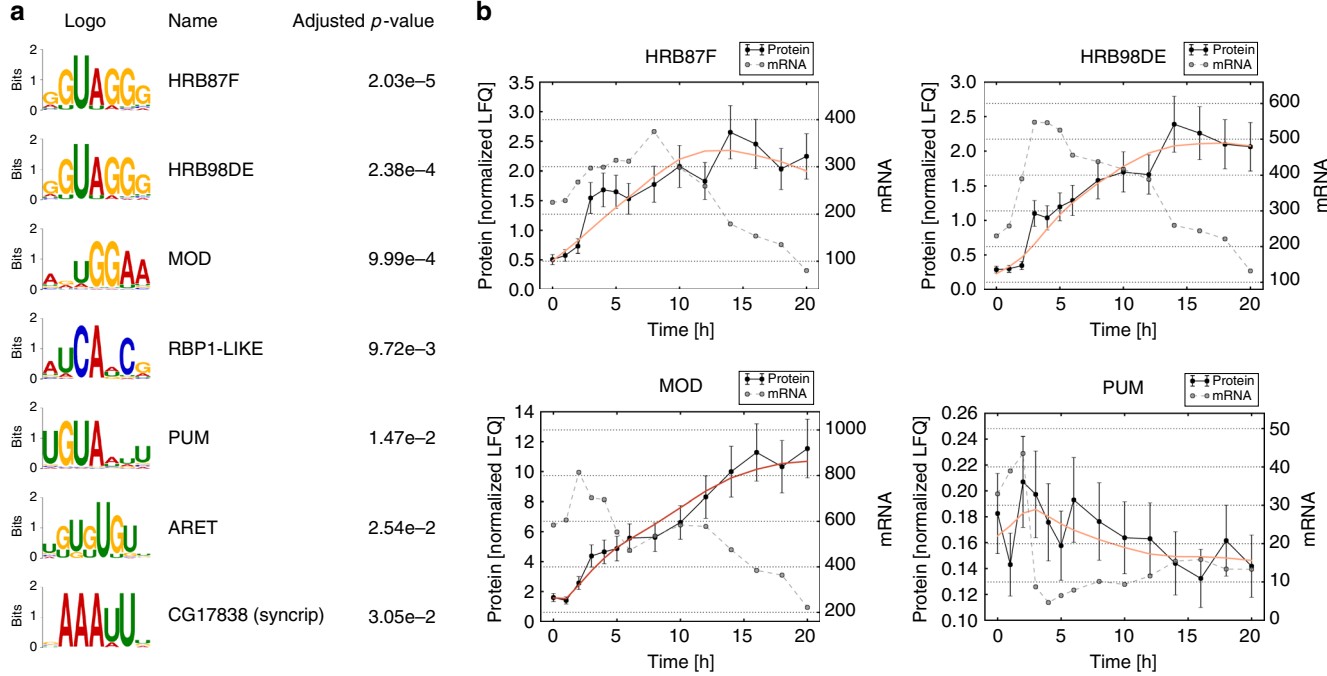

**Fig. 6** Post-transcriptionally regulated proteins are enriched for RBP binding motifs. **a** Seven RBP sequence motifs are enriched in the group of potentially post-transcriptionally regulated proteins (adjusted enrichment $p$ value < 0.05). In total we scanned for enrichment of 67 *Drosophila*-specific motifs corresponding to 51 distinct RBPs. For each RBP, only the sequence logo with highest enrichment is shown. **b** Putative post-transcriptional regulators are regulated at the protein expression level during *Drosophila* development. Measured protein profiles (black solid line—normalized LFQ) of four RBPs, alongside their corresponding mRNA expression (dashed gray line) and the best model fit (colored line) are presented. Error bars represent standard deviation of protein according to a linear error model. The expression level of the three remaining RBPs (RBP1-LIKE, ARET, and CG17838) were below the proteomics detection limit

Hrb98DE in post-transcriptional gene regulation during development, we performed in vivo knockdown experiments with two different dsRNAs (Supplementary Figure 6a) and measured protein expression 7:45 h (29 °C) after egg deposition. The first dsRNA, which reduced Hrb98DE protein by 42% (Supplementary Figure 6a), led to differential expression of 78 proteins of 2600 genes consistently detected in the embryonic time course and the knockdown (3.0%, Supplementary Figure 1b right). In support for our model-based classification approach, there exists a significant overlap between genes predicted to be post-transcriptionally regulated and differentially expressed genes upon knockdown (24 genes observed vs. 13.26 expected, hypergeometric test, $p = 2.6e-3$; Fig. 7a). This significant overlap was preserved in the

knockdown with the second dsRNA, even though knockdown efficiency was lower (Supplementary Figure 6a-c), suggesting that Hrb98DE indeed regulates the predicted targets during *Drosophila* development.

Since assessing the role of Hrb98DE during embryogenesis remains challenging due to limited knockdown efficiency, we turned to *Drosophila* S2R+ cell culture to obtain more robust Hrb98DE depletion. Based on RNA-Seq and mass spectrometry data, we observed a strong depletion of Hrb98DE mRNA and protein by about 84 and 76%, respectively (Supplementary Figure 6d). Off-target effects could be excluded, as two different dsRNA constructs induced highly overlapping transcriptome changes (Supplementary Figure 6e). In the knockdown showing

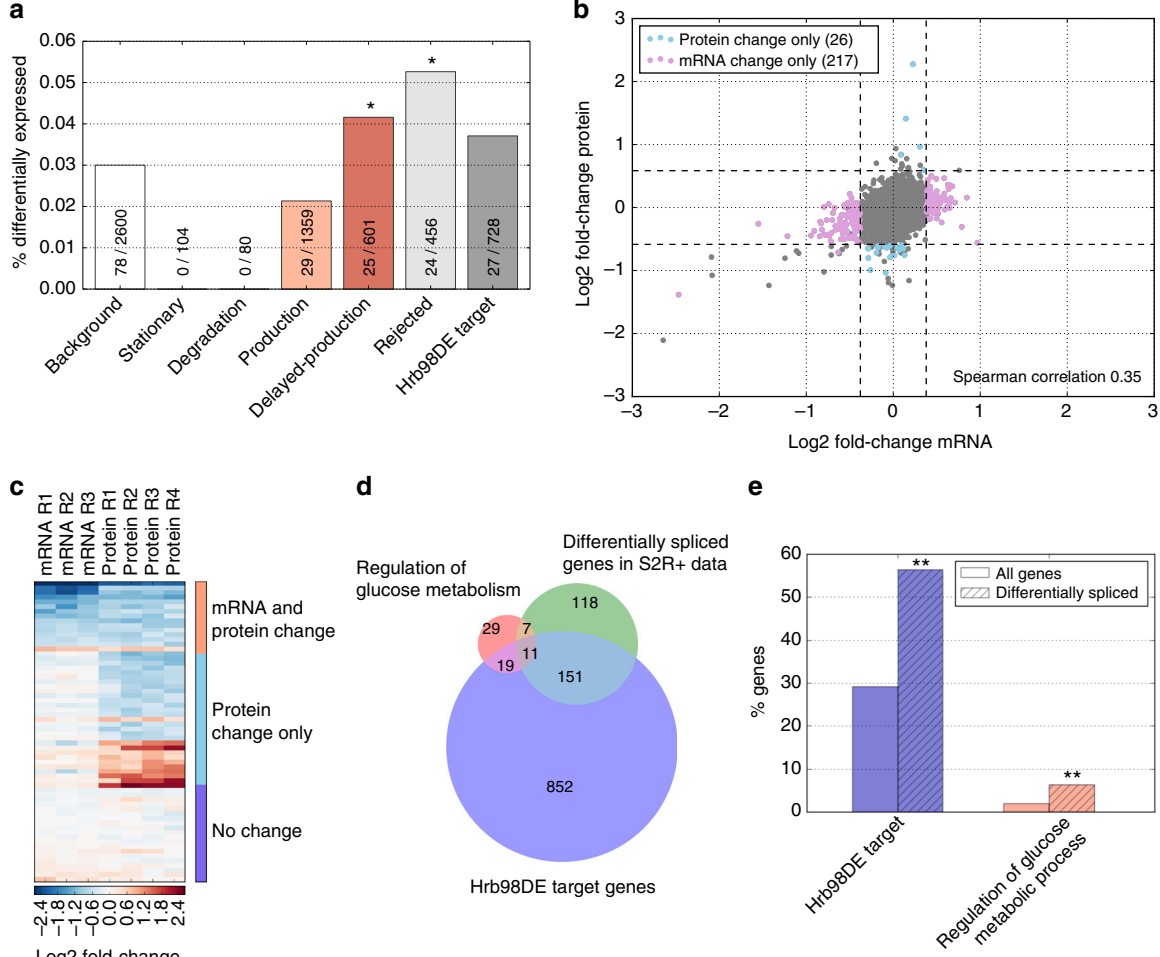

**Fig. 7** Hrb98DE post-transcriptionally regulates glucose metabolism. **a** Proteome changes upon in vivo knockdown of HRB98DE during *Drosophila* development. Bars show the percentage of all genes within each protein class that are differentially expressed upon Hrb98DE knockdown. Absolute numbers are also shown (differentially expressed proteins in class/number of all proteins in class). Significant overrepresentation of differential expression is observed in the rejected and delayed-production categories and preferentially contain an Hrb98DE motif in their mRNA sequence, when compared to the background of all expressed genes (left bar). * indicates $p < 0.05$ using a hypergeometric test. **b** Transcriptome and proteome changes upon knockdown of HRB98DE in S2R+ cells, shown as a scatterplot of mRNA vs. protein fold-changes. Significant cases of only mRNA changing (pink) or only protein changing (light blue) are highlighted. Dashed lines indicate thresholds of fold-changes for mRNA (absolute log2 fold-change >1.3, BH corrected $p$ value < 0.05) or protein (absolute log2 fold-change >1.5, nominal $p$ value < 0.01). **c** Heatmap of fold-changes for mRNA and protein level of differentially expressed genes upon Hrb98DE knockdown in S2R+ cells (in 14 cases mRNA and protein are both significantly changing, in 26 cases only significant change on protein level, 4149 with no change of mRNA or protein). For genes with no significant change in either mRNA or protein only a subset of 20 randomly chosen genes is shown. **d** Splicing changes at the mRNA level upon knockdown of HRB98DE in S2R+ cells. Venn diagram showing differentially spliced genes upon Hrb98DE knockdown and their overlap with genes previously identified as Hrb98DE targets or genes annotated as regulators of glucose metabolic processes (GO:0010906). The previously identified Hrb98DE targets were defined by combining Hrb98DE targets reported by Blanchette et al. (2009), Ji and Tulin (2016)[41] and mRNAs with the Hrb98DE binding motif. **e** Differentially spliced genes are enriched for Hrb98DE targets (blue striped bar) or genes annotated as regulators of glucose metabolic processes (red striped bar), when compared to the background of all detected genes (clear bars). ** denotes significance with a $p$ value below 0.01 (hypergeometric test)

greater Hrb98DE depletion, we detected an overlapping set of 4406 genes at both transcriptome and proteome, out of which 231 are differentially expressed at the mRNA level upon Hrb98DE knockdown (Wald test—DEseq2, BH corrected $p$ value < 0.05, absolute fold-change > 30%), whereas 40 change significantly at the protein level (independent $t$ test two-tailed, $p$ value < 0.01, absolute fold-change > 50%) (Fig. 7b).

Interestingly, a large fraction of genes responding at the protein level (26 out of 40, 65%) do not change at the mRNA level (Fig. 7c), which agrees with published data showing that Hrb98DE binding to mRNA leads only to few instances of changed transcript levels[40]. Thus, we hypothesize that Hrb98DE binds these 26 RNAs to affect protein translation, but not mRNA turnover. In line with Hrb98DE controlling sugar metabolism at the post-transcriptional level, three out of the 26 candidate genes (14-3-3zeta, dorsal, domino) are known to affect glucose metabolism[39].

Hrb98DE has been described as a regulator of alternative splicing[41,42]. To identify additional targets of Hrb98DE-dependent post-transcriptional regulation, we analyzed splicing changes upon knockdown. We focused on 3542 transcripts consistently detected in the S2R+ cell line and the embryonic time course. We found 287 differentially spliced genes upon Hrb98DE knockdown (Fig. 7d), which significantly overlap with the set of direct Hrb98DE targets, which either contain a sequence motif or are published RIP-Seq targets[40,41] (1.93-fold increase observed vs. expected, hypergeometric test, $p$ value = 6.0e-24) (Fig. 7e). We further tested for enrichment of model-predicted post-transcriptionally regulated genes in the set of differentially spliced genes, but did not find a significant overlap. The differentially spliced genes, however, are significantly enriched for the GO-term "regulation of glucose metabolism" (18 genes, 3.37-fold enrichment observed vs. expected, hypergeometric test, $p$ = 2.6e-6), and 11 genes in this set have previously been identified as Hrb98DE targets. Notably, Hrb98DE-dependent alternative splicing of domino, one of the genes involved in glucose metabolism, led to a detectable isoform switch at the protein level (Supplementary Figure 6f).

Our analysis thus suggests post-transcriptional regulation of glucose metabolism by the RBP Hrb98DE, the majority of effects being visible at the level of pre-mRNA splicing.

## Discussion

Early Drosophila development occurs in the absence of de novo transcription, suggesting extensive post-transcriptional gene regulation. In this study, we set out to quantify how mRNA and protein levels are dynamically coordinated, and combined our Drosophila embryogenesis proteome of 14 time points with its paired transcriptome. The data show high biological and technical reproducibility among the four quadruplicates for proteome and transcriptome. This creates an extensive systems-level dataset to study translational and post-translational gene regulation in a highly resolved temporal fashion.

We expressed mRNA and protein concentrations as relative units after normalization by the total amount of mRNA and protein at each time point, as was done in other dynamic transcriptome studies analyzing early Drosophila development[28,43]. This implicitly assumes that total mRNA and protein mass do not change significantly during development. The conclusion of a stable total protein expression across developmental stages is supported by a recent single-embryo proteome survey[44]. Furthermore, a recent study used spike-in controls for absolute quantification of 68 mRNAs in developing Drosophila embryos[45]. We quantitatively compared our relative RNA expression values to their corresponding absolute transcript counts and found a good agreement in time-dependent transcriptome changes (Supplementary Figure 7).

There is a recent interest in explaining the moderate mRNA−protein correlation[3]. While it is appealing to attribute the lack of correlation to a widespread post-transcriptional regulatory network, correlation measures—although practical—do not reflect the complex temporal connection between mRNA and protein dynamics. This is especially true if the correlation analysis is limited to the global mRNA−protein relationship at a single time point. Using mathematical models, we show that a low correlation of mRNA and protein time courses does not necessarily imply post-transcriptional regulation, but could be explained based on protein turnover rates or deviations from steady-state at the onset of development. Thus, the analysis of time-resolved mRNA and protein expression data using mechanistic mathematical models of translation is superior when compared to simple correlation analysis.

Few studies have quantitatively modeled time-course data to investigate the transcriptome/proteome relationship[23–26]. Most of these studies employ protein expression models comparable to ours, but do not explicitly identify post-transcriptionally regulated genes. Using our combined model rejection and model selection approach, we determined post-transcriptional regulation during Drosophila embryogenesis for about 16% of expressed proteins. We establish further confidence in these candidates using a combination of in silico benchmarking and published evidence for post-transcriptional regulation provided by ribosome profiling[21]. We find that post-transcriptionally regulated proteins are enriched for certain biological functions and identify potential mechanisms by RBP motif analysis.

Previous studies also reported comparatively low numbers of post-transcriptionally controlled genes: Investigating the response of dendritic cells to LPS stimulation, Jovanovic et al. can explain 79% of the protein variance based on a simple model of protein translation[26]. They further concluded that post-transcriptional regulation is relevant predominantly for the regulation of absolute protein levels rather than fold-changes upon stimulation. Another study characterized early Xenopus embryonic development and classified proteins into four simple models of translation using model fitting and selection[25]. Only few proteins are not classified into one of the four groups, suggesting limited importance of post-transcriptional regulation during early embryonic development. Two proteomic studies on mammalian circadian rythmicity and on yeast amino acid starvation report fractions of 20−30% of potentially post-transcriptionally regulated proteins[5,29]. This indicates that the low degree of post-transcriptional regulation may be conserved across organisms.

For Drosophila development, our analysis shows that most post-transcriptional control occurs within the first few hours after fertilization, i.e. before or during MZT. In addition, based on our models, we evidenced wide-spread delay of protein production until the onset of MZT. Translation-on-demand mechanisms have been postulated in yeast[31,46] and neurons[47] but to our knowledge not yet for metazoan embryonic development. Interestingly, the proteins with delayed production are enriched for GO terms related to protein catabolic processes, suggesting that induced protein translation may in turn trigger the degradation of maternal proteins at the onset of MZT[35].

Based on sequence motif analyses, we hypothesized that the RBP Hrb98DE is a post-transcriptional regulator of sugar metabolism during early embryogenesis. Indeed, a global remodeling of energy production can be observed during Drosophila development[48]. Collectively, our results and previous work suggest that this remodeling is largely coordinated at the post-transcriptional level[24,29].

Overall, we provide a framework for investigating gene regulation in large-scale paired RNA−protein datasets, by which we identified genes under strong post-transcriptional control.

## Methods

**Collection of embryos for RNA-Seq**. Population cages of wild-type Oregon R flies containing only fertilized females were maintained at 25 °C. Embryos were collected on standard agar apple juice plates in 30 min laying time windows and processed immediately (0 h time point) or aged at 25 °C for the required period (1, 2, 3, 4, 5, 6, 8, 10, 12, 14, 16, 18, 20 h). After collection, embryos were dechorionated using 7.5% hypochlorite for 2 min and rinsed with water. At this point, approximately 30% of the embryos (20 μl embryo pellets) were transferred to PBS buffer for lysis and mass spectrometry measurement. To check for correct and homogeneity of stages, approximately 10% of each sample was fixed and staged. The remaining samples were snap-frozen in liquid nitrogen and stored at −80 °C. Proteome extraction was performed previously as described in Casas-Vila et al.[27]. Total RNA was extracted from approximately 20 μl embryo pellets with the RNeasy Mini Kit (Qiagen) and RNA integrity checked by Bioanalyzer.

**In vivo knockdown experiments**. Females carrying one of two *Hrb98DE* dsRNA transgenes (obtained from Bloomington *Drosophila* Stock Center: # 31303 and 32351) were crossed with males carrying the *actin*-GAL4 driver for 3 days to allow mating. On the fourth day, flies were transferred to conical flasks covered with apple agar plates and females were allowed to lay eggs for 45 min at 29 °C. Embryos were developed for another 7 h at 29 °C and subsequently transferred into tubes containing 1× PBS. For control, actin-Gal4 males were crossed with WT females and experimental flies were harvested and treated in parallel to control flies.

**Cell culture**. Experimental knockdown of Hrb98DE in S2R+ cells was performed using two distinct dsRNAs. *Drosophila* S2R+ cells were cultured at 25 °C in *Schneider's Drosophila* Medium (GIBCO, Cat-No 21720) supplemented with 10% FBS and 2% penicillin/streptomycin. For knockdown experiments, dsRNA was synthesized overnight at 37 °C using the Hi-Scribe T7 kit (NEB, Cat-No-E2040). dsRNA was transfected in S2R+ cells by serum starvation for 6 h. The treatment was repeated twice and cells were harvested 5 days after the first treatment. Primer sequences to amplify dsRNA templates are listed in Supplementary Table 1.

**qRT-PCR**. Three micrograms total RNA was transcribed into cDNA using MMLV reverse transcriptase (Promega). qRT-PCR analysis was performed using a ViiA7 real-time PCR system (Applied Biosystems). Measurements were done in triplicates and relative RNA levels were normalized to *rpl15* levels. qRT-PCR primer sequences are listed in Supplementary Table 1.

**Mass spectrometry measurement and label-free analysis**. For mass spectrometry measurement of Hrb98DE in vivo knockdown, experiments were performed in biological triplicates. Embryos were homogenized in PBS with a microtube pestle, cells were pelleted at 1000 × *g* for 5 min at 4 °C and resuspended in 1× LDS buffer complemented with 0.1 M DTT. Samples were boiled for 10 min at 80 °C and proteins were separated on a 4−12% NuPAGE Bis/Tris gel for 10 min at 180 V in MOPS buffer.

For mass spectrometry measurement of Hrb98DE knockdown in S2R+ cells, experiments were performed in biological quadruplicates. Cell pellets were lysed in RIPA buffer (150 mM NaCl, 50 mM Tris-HCl pH 7.5, 0.1% sodium deoxycholate, 1% igepal CA-630) supplemented with protease inhibitor mix (Roche) for 30 min on ice and vortexed in between. After centrifugation at max speed for 10 min at 4 °C, the cleared supernatant was recovered and protein concentration determined using Bradford. Sixty microgram protein lysate was fractionated by SDS PAGE (three slices per sample).

For Hrb98DE knockdown experiments in vivo and S2R+ cells, gels were cut and destained with 50% EtOH/25 mM ammonium bicarbonate (pH 8) (ABC). After dehydration of the gel pieces with 100% acetonitrile (ACN), samples were dried for 5 min in a concentrator (Eppendorf) and afterwards incubated with reduction buffer (10 mM DTT in 50 mM ABC) for 30 min. The reduction buffer was removed, substituted with alkylation buffer (50 mM IAA in 50 mM ABC) and then subjected to 30 min incubation. Gel pieces were completely dehydrated with ACN and covered in trypsin solution (1 μg trypsin in 50 mM ABC per sample). Proteins were digested overnight at 37 °C. Tryptic peptides were extracted twice by incubation with extraction buffer (3% TFA and 30% ACN) for 15 min and afterwards with 100% ACN. After reduction of the volume of the elution fraction to about 10−20% in a concentrator (Eppendorf), the peptides were passed through a StageTip. StageTips were prepared using two layers of C18 material (Empore) which was activated with methanol, washed with buffer B (80% ACN, 0.1% formic acid) and equilibrated once with buffer A (50 mM ABC, 0.1% formic acid). Extracted peptides were loaded on the StageTips, washed with buffer A and peptides were eluted with 30 μl buffer B and concentrated. Peptides were separated by nanoflow liquid chromatography on an EASY-nLC 1000 system (Thermo) coupled to a Q Exactive Plus mass spectrometer (Thermo). Separation was achieved by a 25 cm capillary (New Objective) packed in-house with ReproSil-Pur

C18-AQ 1.9 μm resin (Dr. Maisch). The column was mounted on an Easy Flex Nano Source and temperature controlled by a column oven (Sonation) at 40 °C using SprayQC. Peptides were separated chromatographically by a 240 min gradient from 2 to 40% acetonitrile in 0.5% formic acid with a flow rate of 200 nl/min. Spray voltage was set between 2.4 and 2.6 kV. The instrument was operated in data-dependent mode performing a top10 MS/MS per MS full scan. Isotope patterns with unassigned and charge state 1 were excluded. MS scans were conducted with 70,000 and MS/MS scans with 17,500 resolution. The raw measurement files were analyzed with MaxQuant 1.5.2.8 standard settings except LFQ quantitation[30] and match between runs option was activated. The standard search parameters were as follows: carbamidomethylation on cysteine as fixed modification, methionine oxidation and protein N-terminal acetylation as variable modification, a minimum peptide length of seven amino acids, specific digestion with trypsin/P with maximal two miscleavages, MS search tolerance for initial search set to 20 ppm and 4.5 ppm for main search, identifications FDR-controlled (<0.01) on peptide level and protein level. Alternatively, quantification was performed using iBAQ[4]. The raw data were searched against the translated ENSEMBL transcript databases (release 79) of *D. melanogaster* (30,362 translated entries) and the *Saccharomyces cerevisiae* protein database (6692 entries). Known contaminants, protein groups only identified by site and reverse hits of the MaxQuant results were removed. A distribution calculated via the logspline R package of each replicate per time point as density function was used to impute the missing values. The mean of measured replicates or the average of two surrounding time points were used as a central value for the imputation distribution calculated using the zoo R package[49]. In case the gap was bigger than a single time point, as well as single measurements with no surrounding values, they were replaced by a fixed small value of 22.5 in log2 scale.

**RNA-Sequencing**. NGS library prep was performed with Illumina's TruSeq stranded mRNA LT Sample Prep Kit following Illumina's standard protocol (Part #15031047 Rev. E). Libraries were prepared with a starting amount of 500 ng and amplified in 11 PCR cycles. Libraries were profiled in a High Sensitivity DNA on a 2100 Bioanalyzer (Agilent technologies) and quantified using the Qubit dsDNA HS Assay Kit, in a Qubit 2.0 Fluorometer (Life technologies). All 68 embryo samples were pooled in equimolar ratio and sequenced on 8 HiSeq 2500 lanes, SR for 1× 51 cycles plus 7 cycles for the index read. S2R+ samples were sequenced with a Mid Output kit using PE2x 79 bp.

The RNA-Seq measurement of the embryo time course yielded an average 18 M reads per sample. Reads were mapped to the BDGP6 fly reference from Ensembl version 79[50] using STAR[51] version 2.4.0 h, allowing up to two mismatches, a minimum intron length of 21, discarding reads mapping to more than ten loci, and eventually keeping only the primary alignment. We assessed the quality of the sequenced reads with FastQC[52], dupRadar[53] and other in-house developed tools. We then counted reads per gene using htseq-count[54] from the HTSeq package with the default "union" mode and using the gene model provided by Ensembl for the same assembly version (BDGP6 version 79).

RNA-Seq of Hrb98DE knockdown in S2R+ cells was performed in biological quadruplicates for each dsRNA. In the S2R+ samples, we obtained an average of 31 M paired reads per sample, assessing the quality using the same strategy described above. For these samples, we used STAR version 2.5.1b to align the reads to the BDGP6 fly reference from Ensembl version 90, using the same parameters as before. To quantify read counts we used an equivalent quantification procedure as described for the in vivo data, applied using the same parameters: FeatureCounts[55] from the Subread package version 1.4.6-p2 was used in order to count reads per gene, also with the default "union" parameters and using the gene model provided by Ensembl for the same assembly version used for mapping (BDGP6 version 90). We estimated the isoform abundance with the Miso package[56], as described in their pipeline (miso --run, miso_pack, summarize_miso). MISO estimates differential splicing of individual exons. For genes containing multiple exons, we designate the complete locus as being differentially spliced if at least one of the exons changes significantly upon knockdown.

**Time-course clustering**. From the mass spectrometry data, we selected 3761 proteins for further analysis based on their number of missing values (maximum four LFQ values below the detection limit out of 14 time points). Both mRNA and protein data including imputed values (see Casas-Vila et al.[27]) was normalized by the respective value at 0 h, log10 transformed, and scaled to minimum and maximum values between −1 and 1. A hierarchical clustering approach was applied to the processed data with imputed values: For this, pairwise distances between all mRNA−protein time courses were calculated using the Euclidean distance. Clusters were merged using a complete-linkage criterion.

**Correlation analysis**. Global Spearman correlation coefficients between mRNA (RPKM) and protein (LFQ, IBAQ) samples were calculated using only proteins showing largest absolute fold-changes compared to the 0 h time point (top 500). Values for which no protein measurement value exists, were not considered in this as well as the following analysis, i.e. no imputed values were taken into account.

We calculated the Spearman correlation coefficients between individual mRNA (RPKM) and protein for all (LFQ: 3761, IBAQ: 4179) time courses based on even time points only ($t = \{0, 2, 4, 6, 8, 10, 12, 14, 16, 18, 20\}$). Shifts were introduced by

matching mRNA at time $t_n$ with protein at time $t_{n+i}$ with $i \in \mathbb{Z} : -5 \leq i \leq 5$. Unmatched time points were left out, leading to a minimum number of six paired values for the assessment of correlation. mRNA/protein correlations with a $p$ value $< 0.05$ ($p$ value returned by scipy.stats.spearmanr—Student's $t$ test, two-sided) were chosen as significant.

**Model fitting and evaluation.** Before fitting, protein data were rescaled by its global mean value over all proteins to avoid numerical issues due to overly large LFQ values. On each time interval between two subsequent measured time points the model variants (stationary, degradation, production, delayed-production) can be solved analytically if the input mRNA time course on each interval is described using a linear function $u(t) = mt + b$. The explicit solution for the delay model

$$\frac{dy(t, \theta, u)}{dt} = \alpha h(t - \tau)(mt + b) - \lambda y(t, \theta, u) \qquad (1)$$

with initial value

$$y(t = 0) = y_0 \qquad (2)$$

then becomes

$$y(t) = \alpha h(t - \tau)\left(b\lambda^{-1} - m\lambda^{-2} + mt\lambda^{-1}\right) + ce^{\lambda t} \qquad (3)$$

with

$$c = y_0 - \alpha h(-\tau)\left(b\lambda^{-1} + m\lambda^{-2}\right). \qquad (4)$$

In the above equation $y$ denotes measured protein, $\alpha$ and $\lambda$ the production and degradation rate, respectively. The Heaviside function $h$ is used to suppress the production term in the delay model with $\tau$ being the time delay.

Model parameters $\theta = \{y_0, \lambda, \alpha, \tau\}$ were estimated by minimizing the weighted least-squares distance between modeled values (Eq. 3) and nonimputed protein data using a trust region reflective optimization scheme (scipy.optimize. least_squares in Python). Missing protein values (or their imputed counterparts) were not considered in the cost function.

$$\chi^2(\theta, u) = \sum_i^n \frac{\left(y(\theta, u)_i^{\text{model}} - y_i^{\text{data}}\right)^2}{\sigma_i^2}. \qquad (5)$$

Here $n$ denotes the number of data-points, $y$ modeled or measured protein expression, $\theta = \{y0, \alpha, \lambda, \tau\}$ the parameter vector, and $\sigma$ the weight for each data-point $i$. By relating mean and standard deviation of the four biological replicates, a linear error model was chosen to express $\sigma$ dependent on mean protein expression (slope, 0.169, intercept, 0.0). Ranges for the parameters are chosen as follows: $y_0 \in [1e-1, 5e3]$, $\lambda \in [\ln(2)/1e3, \ln(2)/1e-1]$, $\alpha \in [1e-5, 1e-1]$, $\tau \in [1e-5, 1e1]$. In all cases, multi-start local optimization was carried out via latin-hypercube sampling and parameters were sampled on a logarithmic scale. Depending on the model complexity, a different number of initial parameter samples was chosen (degradation model: 5, production model: 5, delayed-production model: 25).

Model rejection was carried out by applying a $\chi^2$-test to the weighted squared residuals between model and data[57]. To further exclude strong systematic deviations between model and data, the residuals were subjected to a Durbin−Watson (DW) test. Normality of model residuals was checked comparing residuals of the complete model (delayed-production) with a standard normal distribution (Supplementary Figure 8). The empirical distribution of both test statistics ($\chi^2$ and DW-test) was determined using a parametric bootstrap approach. Here the best model fit was resampled 1000 times assuming normally distributed noise with mean zero and standard deviation corresponding to the data-point of the randomized observable. Models were fit to bootstrapped data using the optimization procedure outlined above with estimated parameters of the original fit as initial parameter vector. $p$ values for each test statistic were calculated from the empirical cumulative density function of their distribution and corrected via the Benjamini−Hochberg (BH) procedure. Only if a model fit passed both $\chi^2$- and Durbin−Watson-tests (BH corrected $p$ value < 0.05), a model variant was considered feasible for the mRNA−protein pair under consideration.

If multiple models remain possible for a given mRNA−protein pair after testing, model selection was carried out using a stepwise likelihood-ratio test with $\alpha = 0.95$ (one-tailed). For both the stationary and the degradation model, correction criteria were applied according to the procedure described in Supplementary Note 3. For all estimated parameters 95% confidence intervals were calculated using a profile likelihood approach[58] (see Supplementary Data 1).

**Model analysis.** Proteins with an estimated upper confidence interval limit of the half-life below the median value of all protein half-lives were selected as proteins with short half-lives (313 proteins). Inversely, the lower confidence interval limit needed to be above the median estimated half-life for a protein to be classified as having a long half-life (237 proteins).

Theoretical protein steady-states were calculated based on estimated parameter values as

$$y_t^{\text{ss}} = \alpha u(t)/\lambda, \qquad (6)$$

where $\alpha$ denotes the estimated protein production rate, $\lambda$ the estimated protein degradation rate and $\mathbf{u}(t)$ the measured mRNA abundance either at $t = 0$ or $t = 20$. A protein was considered in steady-state at $t = 0$ if its measured concentration at this time point was within 20% of the estimated steady-state at $t = 0$. This resulted in 481 proteins to be classified as in steady-state, while a combination of this set with proteins with short half-lives consisted of 95 proteins. Correlation was calculated as in section Correlation analysis of the Methods with no time shift considered between mRNA and protein.

**Gene set enrichment analysis.** GO term enrichment for protein groups lists was carried out using Gorilla[59]. As background the list of 3761 reproducibly measured proteins was chosen. Results were filtered using a corrected $p$ value of <0.05 (Benjamini and Hochberg correction). Plotting was done using Multi-Dimensional Scaling implemented in the sklearn package provided for python based on a distance metric obtained using semantic similarity between GO terms. Semantic similarities were retrieved from the bioconductor package "GOSemSim"[60] using the Wang distance metric.

Seventeen proteins with the following GO terms have been selected for Supplementary Figure 5a: "monosaccharide metabolic process" (GO:0005996), "glucose metabolic process" (GO:0006006), "monosaccharide biosynthetic process" (GO:0046364), "carbohydrate biosynthetic process" (GO:0016051), "hexose metabolic process" (GO:0019318), "gluconeogenesis" (GO:0006094), "hexose biosynthetic process" (GO:0019319). Proteins (34) assigned the following GO terms are plotted in Supplementary Figure 5b: "mitotic spindle organization" (GO:0007052), "mitotic cell cycle process" (GO:1903047), "cell cycle process" (GO:0022402), "microtubule cytoskeleton organization involved in mitosis" (GO:1902850), "mitotic centrosome separation" (GO:0007100).

**Motif analysis.** Motif analysis was carried out using AME[61]. Input sequences were defined as one of the following: the common transcript sequence between all mapped transcripts for each protein, the longest transcript sequence, longest 3′ UTR sequence, longest 5′UTR sequence, or longest coding sequence. The motif database was taken from Ray et al., which included 67 motifs mapping to 51 RBPs[37]. Using AME, sequence motif scoring method was based on total hits. Motif match threshold was $p < 0.0002$ (Fisher's exact). Background was estimated from the sequences of the 3761 reproducibly measured proteins. As control all sequences not in the group were chosen. Statistical test for enrichment/association was Fisher's exact. The threshold values for reported results was set to an adjusted $p$ value of 0.05 (Benjamini and Hochberg correction). Individual motif occurrences were identified using FIMO[62] with a $p$ value threshold of $p < 0.0002$ (Fisher's exact) for reported motifs. Multiple motif copies in the motif database were grouped by calculating the union over all transcripts.

**Statistical analysis of Hrb98DE knockdown in vivo.** Only proteins quantified in all measured samples (*actin* KD1 and *actin* KD2, each in biological triplicates) were chosen for statistical analysis of the in vivo knockdown. Additionally only the overlap of genes uniquely identified in the embryonic time course as well as in the in vivo knockdown were considered (2600 genes). Differentially expressed proteins were determined using an independent $t$ test ($p < 0.01$, two-tailed) and absolute fold-change > 2.0. Significant overlap between differentially expressed genes and different model classes or Hrb98DE targets was assessed using a hypergeometric test.

Genes are considered targeted by Hrb98DE if identified as such by the study of Ji and Tulin[40], the study of Blanchette et al.[41] or containing an Hbr98DE binding motif (section Motif analysis). Target genes identified in Ji and Tulin were kindly provided by Alexei Tulin. Target genes according to Blanchette et al. were determined re-analyzing the raw data according to the methods described in ref. [57], and using a fold-change cutoff of log2 > 10 between immunoprecipitation and control samples.

**Statistical analysis of Hrb98DE knockdown in S2R+ cells.** Protein with NaN values in any of the tested conditions were filtered out before differential expression was assessed using an independent $t$ test between Hrb98DE knockdown and LacZ control. As thresholds for differentially expressed protein a $p$ value < 0.01 ($t$ test, two-tailed, $n = 4$) and fold-change cutoff > 50% was chosen. In the RNAseq data, nonexpressed mRNAs were removed based on a mean RPKM over all samples > 1. We obtained mRNA statistics using DESeq2[63] version 1.14.1, using the default Wald test to calculate the significance and applying automatic independent filtering to avoid testing genes which were poor candidates of being differentially expressed (maximizes the number of adjusted $p$ values less than alpha = 0.1). Differentially expressed mRNA were identified using a threshold of <0.05 for BH corrected $p$ values ($n = 3$) and a fold-change cutoff of >30% change.

To identify differentially spliced genes we used DEXSeq[64] in the following way: we recounted reads on exons using the default htseq union mode; not discarding reads spanning multiple exons; taking into account that data is paired-end but

without discarding singletons. DEXSeq is a specialization of the NB-GLM approach and internally uses DESeq2 to test for differential exon usage, and we modeled our data with the formula "~ sample + exon + condition:exon", looking for changes in exon abundance that could be explained by the KD (hence the interaction term in the formula). Finally, we considered that genes could show a differential splicing pattern if they included exon abundance changes of at least 50% between conditions at FDR 1%. Dupradar, DESeq2 and DEXSeq are part of the Bioconductor[65] project. Genes associated with the GO-term "regulation of glucose metabolic process" (GO:010906) were retrieved from FlyMine v45.1 2017.

**Code availability**. The code to classify paired mRNA/protein time courses is available using the following link: https://github.com/Legewie/PyProt (https://doi.org/10.5281/zenodo.1435819). All code made available is licensed under the GNU General Public License (Version 3, 29 June 2007).

## Data availability

The mass spectrometry proteomics data have been deposited to the ProteomeXchange Consortium via the PRIDE partner repository with the dataset identifier PXD011238. The RNA-Seq data of this publication have been deposited in NCBI's Gene Expression Omnibus and are accessible through GEO Series accession numbers GSE121160 and GSE121161. A Reporting Summary for this Article is available as a Supplementary Information file. All other data supporting the findings of this study are available from the corresponding authors on reasonable request.

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

## Acknowledgements

We acknowledge the help of Hannah Lukas and Maria Méndez-Lago (IMB Genomics Core Facility) for RNA sequencing. We thank Alexej Tulin for kindly providing a list of Hrb98DE targets, Stefanie Ebersberger for valuable advice on motif analysis and Bertrand Fabre for discussion on protein normalization. S.L. and K.B. were supported by the e:bio junior group program of the German Federal Ministry of Education and Research (BMBF, FKZ: 0316196) and by the German Research Foundation (grant LE 3473/2-1). Part of the work was supported by the Rhineland-Palatinate Forschungsschwerpunkt GeneRED.

## Author contributions

A.B. and N.C.-V. carried out embryo collection and preparation of samples for proteome and RNA-seq measurement. Processing of raw protein data was carried out by M.D. Raw reads from RNA-seq were processed by S.S. S.S. and K.B. analyzed correlation between data from Graveley et al. (2011) and our RNA-seq data. K.B. carried out time-course clustering, correlation analysis of transcriptome and proteome, model fitting and selection, model analysis, gene set enrichment analysis, and motif analysis. C.K. provided input concerning model fitting and selection procedures. N.D. prepared *Hrb98DE* knockdown cell lines and prepared samples for RNA-sequencing. N.C.-V. and A.B. prepared *Hbr98DE* knockdown cell-line samples for proteome measurement. Statistical analysis of *Hrb98DE* knockdown results was done by K.B. while analysis of differential splicing was carried out by S.S. K.B., F.B., S.L., and J.-Y.R. designed the project. All authors discussed the results. K.B., F.B., and S.L. wrote the manuscript with input from all other authors.

## Additional information

**Competing interests:** S.S. is currently employed at Boehringer Ingelheim International GmbH. The remaining authors declare no competing interests.

