## [Peer Review File · Nature Communications]

Reviewers' Comments:

Reviewer #1:

Remarks to the Author:

In this manuscript, Becker et al. investigate the relationship between mRNA and protein abundance during the highly dynamic process of *D. melanogaster* embryo development. The authors take advantage of a proteomic time-course dataset recently published by them (Casas-Vila, et al., Genome Research 2017) and integrate it with RNAseq data obtained from the same samples. Similar RNAseq dataset were previously published by others, but the authors argue that having proteomic and transcriptomic data from the same samples might improve modeling. The authors confirm previously reported (also by them) low level of correlation between mRNA level and protein abundance during fly embryo development, and then demonstrate that by using a limited set of Ordinary Differential Equations (ODEs) it is possible to model dynamics of protein and transcript abundances for up to 80% of the quantified cases. Finally, the authors point to additional post-transcriptional mechanisms not taken into account in the models to explain the remaining 20% of the cases. In particular, they find an enrichment of motifs recognized by RNA binding proteins. Among these, they select the RNA binding protein HRB98D and attempt to demonstrate a potential role for these proteins in regulating glucose metabolism during embryo development via post-transcriptional mechanisms. The manuscript is well written and easy to follow, and the underlying data appear of high quality (even though in part already published). The major novelty derives from the modeling with ODEs, which appears logically well designed, even though I am not a modeling expert. The weakest point of the manuscript is the functional validation of the role of HRB98D in regulating glucose metabolism. In particular, I have serious doubts that the experiment performed support the claim made by the authors. Unless these functional aspects are considerably strengthened and supported by *in vivo* experiments, I find this manuscript more appropriate for a more specialized, "systems biology" oriented journal.

Major points:

- The authors should study *in vivo* the role of HRB98D during embryo development. They could use mutants or knock-down of HRB98D using germline specific lines and address whether this perturbation affects embryo development. If HRB98D has a relevant role in regulating glucose metabolism, I would expect some kind of effect on embryo development. Additionally, they should repeat the proteomic and RNAseq measurements upon perturbation of HRB98D to demonstrate its influence on protein abundance by post-transcriptional mechanisms. In this case, perhaps not all the time course need to be repeated, but select time points could be chosen on the basis of the experiments already performed.
- It is not clear to me what is the rationale behind checking for alternative splicing events in S2R+ cell lines upon siRNA of HRB98D (Figure 7D). How is this supposed to influence the relationship between mRNA and transcript level? Do alternative spliced transcript show also protein abundance changes? Are the same splicing events also detected in the *in vivo* data?
- There is no description of how the siRNA experiments were performed. How many siRNA were used? If only one, how about off-targets effects?
- It is not clear to me how the correlation between protein and mRNA levels were calculated. Which values were used in Fig. 2B? How were LFQ values "normalized"? Unless these were corrected for protein size, I found it a bizarre choice of using LFQ instead of iBAQ values. LFQ are generally not normalized by protein size while RPKM are (reads per kilobase). Using not normalized data for protein levels could of course reduce the correlation between mRNA and protein levels.
- Were the RNAseq experiments performed using Spike-In controls? Burst in transcription at the end of MZD could lead to all transcripts being suddenly up regulated and normalization might produce artificial fold changes upon normalization. The authors should at least comment on this.

Minor points:

- Some of the mass spec data were imputed, have the authors assessed the impact of the correlation between protein and mRNA with and without imputation?
- For the siRNA experiment, what is the rationale for choosing cut offs for RNAseq and proteome experiments? Is the cut off for proteomics data really based on "p value" and not adjusted for multiple testing? If this is case, it needs to be fixed.
- While the manuscript is clearly written the Figures need to be improved. Some figures are cropped at the bottom (e.g. Fig.2B, Fig.4), some axis-label missing (Fig. 5).

Reviewer #2:

Remarks to the Author:

General remarks

The authors have investigated the mRNA-protein relationship during *Drosophila* embryogenesis using matching transcriptome and proteome data for 14 developmental time points. They observe that mRNA-protein correlations improve when considering delayed time points and that further improvements can be obtained using mass-action type models of translation and protein degradation. A notable discrepancy with a simple mass action model for a significant fraction of genes is due to the observed maternal-to-zygotic transition (MZT), after which the mass action models capture well the protein-mRNA relation. Finally, the authors have stated that proteins belonging to different kinetic models are enriched for different GO terms and their analysis has revealed RNA-binding protein Hrb98DE to be a post-transcriptional regulator of genes involved in sugar metabolism, which was validated by follow-up experiments.

Overall, this study did not delineate well its novelty.

Indeed, the biological conclusions of the authors about poor mRNA-protein correlations and kinetic models providing better insights about their relationship have been reported before (Jovanovic et al. 2015, Peshkin et al. 2015). Hence the most important biological message is already known. Furthermore, Peshkin et al. 2015 had generated and analysed a similar data for *Xenopus* and also reported differences for eth MZT transition. Hence the biological insights are confined to the fly community and to the RNA-binding protein Hrb98DE twist of the paper.

Methodologically, the modeling approach has not been compared against alternative existing approaches (eg Teo et al 2014, Peshkin et al. 2015) nor it has been thoroughly benchmarked against simulations and independent data. Also, the model is based on strong modelling assumptions (absolute vs relative concentrations), and non-assessed assumptions important for statistical testing (normality). They were also errors in the python code. Therefore, the modelling itself will not be likely reused.

Altogether, I recommend rejection of the paper. It would need re-working of the model, a clear delineation of the contribution of this study to state-of-the-art rather than a mere referencing (additional important references are Robles et al PLoS Genet, 2014, Cheng et al Mol Syst Biol 2016 and Lau et al., Nature comms, 2018.), and the retargeting to a more specialized journal. I hope my detailed comments below will help the authors to this end.

Minor points

1. Abstract. "the direct correlation between mRNA levels and protein abundances is moderate in

most studies" Most studies are comparing steady-state conditions. For steady-state data, using ODE would not help. Hence, the argument does not help for "most studies".

2. Abstract. "ODE models describing protein translation and degradation fit 81% of mRNA-protein profiles" is unclear. One needs to read the paper to understand it. I understand that no correction for multiple testing was done (see below).

3. The discussion of the correlation of present RNA-seq data with previously published one is a bit lengthy and off the core topic of this paper. It could be made more succinct.

4. From the publication and website, MISO seems to only estimate isoform distribution of genes with multiple isoforms (PSI). How did they get values for the whole locus?

5. The iBAQ quantification has been developed for the purpose of absolute quantification. The authors should try with iBAQ instead of LFQ and report whether it improves the results.

6. Since the protein and RNA quantifications are relative to each other, the ODEs assume that the total RNA mass and the total protein mass does not change over time. Is that realistic? That assumption should be made explicit and discussed if not assessed. Also RNA-seq protocol depletes for ribosomal RNAs.

7. Why were the SR2+ data analysed differently (read mapping and counting, no isoform distribution estimation)?

8. "In contrast to the Pearson correlation coefficient, Spearman correlation makes no assumptions about normality of the data or linearity in the dependence of the variables (de Sousa Abreu et al., 2009; Tobias Maier et al., 2009)". The ref (de Sousa Abreu et al., 2009; Tobias Maier et al., 2009) are not appropriate for this statement. It is not their work but Spearman's.

9. "We included only significantly changing proteins (n = 866) to avoid that random variations around the mean might mask existing trends between mRNA and protein". What is the statistical test and multiple testing procedure applied? Which significance cutoff was used? The method refers back to Casas-Vila et al. Gen Res 2017. However, i) these informations are important to understand this paper so they must be recapitulated here and ii) Casas-Vila have multiple ways of selecting differentially regulated proteins. I could not find one that had the same number (866) than here.

10. It is not the "heavy-side" function, but the "Heaviside" function, named after its inventor.

11. Equation (3) must be $\exp(-\lambda t)$ and not $\exp(\lambda t)$.

12. Equation 4: $h(-\tau)$ should be $h(t-\tau)$?

13. If Equation (4) is correct (I think it is up to point #12) then the python code line 29 and line 94, which calculates "c" in solve_modelD and in solve_modelC in the file solve_model.py are wrong. This must be corrected and the analysis re-done.

14. The code and Suppl Table 1 should be provided (I got them afterward)

15. According to the code, $u(t) = mt + b$ is assumed in each interval. Hence interval indices should be introduced for the slope and intercept for clarity.

16. How good does this model fit overall to the data? A global scatterplot must be provided.

17. The model is written for protein and RNA expression in the natural scale. However, both for RNA and protein, noise is typically multiplicative in the natural scale. Therefore, the fitting can suffer from heteroscedasticity, whereby high expression data points, which also present largest noise, have strongest influence on the fits. Is that the reasons why weighted least squares are used? If so, how the weights are computed should be made clear.

18. Normality assumption of the residuals is important for the chi-squared test. A quantile-quantile plot of the residual must be provided. If Normality is not met, alternative non-parametric testing strategies must be investigated and applied (eg Bootstrap of cases or of residuals). The same issue applies to the Durbin-Watson test.

19. The cut-off on type I error was based on nominal p-value ($\alpha = 0.05$). The authors should consider multiple testing corrections.

20. The initial value y_0 should be fitted rather than forced to be equal to the first measurement value. The reason is that every measured values entail noise. Errors in y_0 propagate to all data points. They may also be a strong driver of the anti-correlation seen in Fig 4B.

21. Fig 4B should be backed by independent measures. For instance, by comparing half-lives from other measurements of protein half-life and scatter plotting. Also, the anti-correlation might be a statistical artefact (correlation of estimation errors). One way to address this issue is to check that the trend holds when production rates and half-lives are estimated from independent subsets of the data (e.g. using the odd data points for production and the even data points for half-life).

Reviewer #3:

Remarks to the Author:

This paper is a treasure trove of new information that should stimulate much productive further investigation. The most important point - for this reviewer - is the demonstration that the poor correlation between transcript and protein titers can, in a majority of cases, be explained without resort to idiosyncratic (i.e. gene-specific) post-transcriptional regulation. The authors show that relatively simple models containing delays and accounting for protein degradation do the trick. The robustness of the allocation of various genes to these separate models might have been strengthened by additional Monte Carlo simulations, but I consider that a quibble; the particular assignments aren't the great take-away lesson. The success in reconciling the majority of the cases is -- and that result should stimulate much new work by the authors and by others. The HRB98DE results I count as an additional bonus. They are persuasive and, again, I think they will stimulate others to try to do similar things.

The manuscript is well written. A few figure legends could have been clearer, but they can be worked out.

Response to reviewers comments

Reviewer #1 (Remarks to the Author):

In this manuscript, Becker et al. investigate the relationship between mRNA and protein abundance during the highly dynamic process of *D. melanogaster* embryo development. The authors take advantage of a proteomic time-course dataset recently published by them (Casas-Vila, et al., Genome Research 2017) and integrate it with RNAseq data obtained from the same samples. Similar RNAseq dataset were previously published by others, but the authors argue that having proteomic and transcriptomic data from the same samples might improve modeling. The authors confirm previously reported (also by them) low level of correlation between mRNA level and protein abundance during fly embryo development, and then demonstrate that by using a limited set of Ordinary Differential Equations (ODEs) it is possible to model dynamics of protein and transcript abundances for up to 80% of the quantified cases. Finally, the authors point to additional post-transcriptional mechanisms not taken into account in the models to explain the remaining 20% of the cases. In particular, they find an enrichment of motifs recognized by RNA binding proteins. Among these, they select the RNA binding protein HRB98D and attempt to demonstrate a potential role for these proteins in regulating glucose metabolism during embryo development via post-transcriptional mechanisms. The manuscript is well written and easy to follow, and the underlying data appear of high quality (even though in part already published). The major novelty derives from the modeling with ODEs, which appears logically well designed, even though I am not a modeling expert. The weakest point of the manuscript is the functional validation of the role of HRB98D in regulating glucose metabolism. In particular, I have serious doubts that the experiment performed support the claim made by the authors. Unless these functional aspects are considerably strengthened and supported by *in vivo* experiments, I find this manuscript more appropriate for a more specialized, "systems biology" oriented journal.

Major points:

1) The authors should study *in vivo* the role of HRB98D during embryo development. They could use mutants or knock-down of HRB98D using germline specific lines and address whether this perturbation affects embryo development. If HRB98D has a relevant role in regulating glucose metabolism, I would expect some kind of effect on embryo development. Additionally, they should repeat the proteomic and RNAseq

measurements upon perturbation of HRB98D to demonstrate its influence on protein abundance by post-transcriptional mechanisms. In this case, perhaps not all the time course need to be repeated, but select time points could be chosen on the basis of the experiments already performed.

We agree with the reviewer that *in vivo* analysis of the role of Hrb98DE during embryonic development would be the most direct validation of our model predictions. We thus performed the proposed *in vivo* validation experiment. In detail, we performed *in vivo* Hrb98DE knockdown using two different dsRNA constructs, and collected flies 7:45h at 29°C after egg deposition for analysis (corresponding to 10h at the previously used temperature of 25°C). We separately used two different driver lines for the expression of the dsRNAs: an *actin*-GAL4 driver that starts expressing GAL4 at MZT and a *nanos*-GAL4 driver that already expresses in the germline. Each experiment was performed in triplicates, thus resulting in a total of 18 knockdowns (3 dsRNAs x 2 drivers x triplicates). We observed low knockdown efficiencies (~50%) on protein and mRNA levels when KD was generated with the *actin* driver line, as judged by qPCR and mass spectrometry, respectively (Figure 1 and not shown). With the *nanos*-GAL4 driver line, Hrb98DE protein was mostly unchanged compared to animals expressing a non-specific dsRNA, possibly because dsRNA expression was too low when controlled by this driver (Figure 1). Nevertheless, using mass spectrometry, we compared the proteomes of control embryos to the *actin*-GAL4 driver-mediated KD3, showing the greatest knock-down on protein level (40% expression relative to control). Within the overlap of expressed proteins in this data with the 3761 proteins expressed in the embryo time-course (2531 proteins), we found 75 differentially expressed proteins (p-value < 0.01, fold-change > 2.0). Despite the low knock-down efficiency, we observe a significant overlap between this set of significantly changing proteins and transcripts containing the Hrb98DE motif (hypergeometric test, p-value < 0.05). Further there is a significant overlap between significantly changing proteins and proteins predicted to be post-transcriptionally regulated based on our model-based classification (hypergeometric test, p-value < 0.05). The same conclusion holds true when we analyze *actin*-GAL4 driver mediated KD1, which shows reduced knock-down efficiency compared to KD3. Taken together, these results indicate that our modeling approach indeed correctly predicts post-transcriptionally regulated proteins and Hrb98DE is involved in the mechanism of regulation. We have now included these new results in the revised version of the manuscript.

As demonstrated the *in vivo* knock-down of Hrb98DE indicates promising results, but it remains difficult to target Hrb98DE for knockdown, perhaps because the maternally provided Hrb98DE is exceptionally stable, preventing efficient depletion of the protein early in development. The alternative strategy of knocking out Hrb98DE is impossible as loss of Hrb98DE is lethal (Ji and Tulin, 2012). A last option to circumvent this issue would be to generate specific germline KO clones in an otherwise wild type background. However, Hrb98DE was shown to be required for normal oogenesis, which prevents us to obtain and examine early embryos (Ji and Tulin, 2012).

Figure 1 - Protein expression changes upon *in vivo* knock-down of Hrb98DE as assessed by mass spectrometry: The left panel shows the observed residual Hrb98DE protein expression upon knock-down using two different GAL4 driver lines (actin, nanos) in combination with three dsRNAs. The middle panel shows a scatter plot comparing protein expression between actin KD3 and control. Red dots highlight significantly changing proteins. Right panel: Differentially expressed proteins are significantly ($p < 0.05$, indicated by *) enriched for genes containing an post-transcriptionally regulated proteins predicted by the model (middle bar) and Hrb98DE motifs (right bar). The left bar shows the percentage of differentially expressed in the background set of all differentially expressed proteins.

Taken together, the validation of Hrb98DE function in early embryo indicates some promising results, but is compromised by low KD efficiency and the requirement of Hrb98DE during oogenesis. In a personal communication, the editor of *Nature Communications*, therefore, in principle, agreed that a detailed analysis of *in vivo* knock-down of Hrb98DE is not absolutely required in light of these technical challenges. We nevertheless tried the proposed *in vivo* validation and explain in the following why the results were not conclusive. We therefore had to resort to SR2+ cells that are more suitable to confirm the predicted post-transcriptional regulation, as strong knockdown can be achieved without adverse effects on cell survival. We included the *in vivo* knockdown in the revised Results section, while discussing its limitations and still using the more robust knockdown in the SR2+ cell culture system as the main validation for the role of Hrb98DE in post-transcriptional regulation.

2) It is not clear to me what is the rationale behind checking for alternative splicing events in S2R+ cell lines upon siRNA of HRB98D (Figure 7D). How is this supposed to influence the relationship between mRNA and transcript level? Do alternative spliced transcript show also protein abundance changes? Are the same splicing events also detected in the vivo data?

Based on the combination of sequence motif analyses and analysis of functional enrichment, we hypothesized that Hrb98DE may be a post-transcriptional regulator of sugar metabolism.

Post-transcriptional regulators frequently affect multiple aspects of RNA metabolism. For instance, it is well established that micro RNAs affect mRNA stability as well as mRNA translation into protein (Bushati et al. 2008, Valencia-Sanchez et al. 2006). Likewise, Hrb98DE has not only been described as a regulator of mRNA translation, but also as a factor controlling RNA splicing (Blanchette et al. 2009, Brooks et al. 2015, Ji & Tulin 2016). Hence, we decided to analyze knockdown effects on multiple mechanisms of post-transcriptional regulation.

Upon knockdown, we found the predicted evidence for translational regulation, as a large fraction of genes responding at the protein level (26 out of 40) do not show a concomitant change at the mRNA level (Figure 7B and 7C). In line with our hypothesis, some of these putative translationally regulated genes function in sugar metabolism. However, at the level of splicing changes, the Hrb98DE knockdowns were more pronounced (higher number of affected mRNAs), overlapped better with previously reported direct Hrb98DE targets and were highly enriched for the biological function 'regulation of glucose metabolism' (GO:0010906). This suggests that in SR2+ cells, where we perform the knock-down, the majority of Hrb98DE-dependent effects on glucose metabolism are visible at the level of pre-mRNA splicing. In the revised manuscript, we explain this rationale in more detail.

Concerning the relationship between splicing and protein vs. transcript levels, we reported in the original manuscript that splicing changes observed at the RNA level are, in certain cases, also eliciting corresponding isoforms switches in the respective proteins (Fig. S6F). Such observations were, however, limited to a few cases, since only very few splicing changes can be detected in proteomic data due to the requirement of having unique peptides for both isoforms. During the revision, we further investigated whether splicing changes at the RNA level are related to absolute abundance changes at the protein level (sum of all protein isoforms). We found that indeed splicing changes are visible at the level of protein, as 13 out of 40 (32.5%) differentially expressed proteins also show splicing changes. For comparison, the fraction of differentially spliced genes in the considered background of expressed mRNA and protein is merely 5.7%. Therefore, a significant enrichment of differentially spliced mRNA is observed in the group of differentially expressed protein (5.7-fold enrichment, hypergeometric test, $p = 1.5e-7$).

We agree with the reviewer, that it would be of interest to check also for splicing changes in the *in vivo* time-course data. However, time-resolved embryonic RNA sequencing was performed using single-end 50bp sequencing to accommodate the large number of samples. This sequencing mode has high throughput and allows for the quantification of total mRNA per locus, but it is difficult to reliably detect splicing changes. We therefore did not attempt a time-resolved splicing analysis.

3) There is no description of how the siRNA experiments were performed. How many siRNA were used? If only one, how about off-targets effects?

We do agree with the reviewer about the importance of checking for potential off-target effects. We therefore performed RNA-Sequencing measurements also for a second knock-down experiment of Hrb98DE with an independent dsRNA. We see good overlap between both dsRNA as well as a highly significant overlap of differentially expressed mRNA (Figure 2). We therefore conclude that off-target effects are minimal. These results are included in the revised manuscript (Supplemental Figure S6E).

Figure 2 - Independent dsRNA-mediated knockdowns of Hrb98DE in SR2+ cells show highly similar effects at the mRNA level: The left panel shows the Pearson correlation (see colormap) between Hrb98DE knock-downs using two different dsRNA (KD1 and KD2), each being performed in triplicates (e.g., KD1_1 to KD1_3). The knockdown and control replicates are more correlated within than across each other. The right panel shows a strong overlap of the sets of differentially expressed mRNAs (BH corrected p-value < 0.05, fold-change > 1.5) in the two knock-down experiments.

4) It is not clear to me how the correlation between protein and mRNA levels were calculated. Which values were used in Fig. 2B? How were LFQ values “normalized”? Unless these were corrected for protein size, I found it a bizarre choice of using LFQ instead of iBAQ values. LFQ are generally not normalized by protein size while RPKM are (reads per kilobase). Using not normalized data for protein levels could of course reduce the correlation between mRNA and protein levels.

We agree with the reviewer that iBAQ values are normalized for protein size (whereas LFQ values are not), and thus represent a more direct measure of protein concentrations. The choice of LFQ vs. iBAQ may influence the global RNA-protein correlation (all genes - single time point; Fig. 2A), but to our knowledge no systematic comparison how this affects modeling has been done. For the protein time course, the two measures are expected to mainly differ by a scaling factor, suggesting that the local RNA-protein correlation (one gene - all time points; Fig. 2C) will be essentially unaffected using iBAQ, as will be the modeling results relating RNA protein time courses (except for a change in the estimated protein translation rate). Thus, following our original publication for the proteome dataset (Casas-Vila et al, 2017), which was reported as LFQ values we stuck to modeling with the LFQ values.

To address the reviewers' comment (also raised by reviewer #2, minor point 5), we re-

calculated the global RNA-protein correlation between samples, and indeed found a slightly better correlation using IBAQ values. However, the RNA-protein correlation still remained comparatively low, the maximum Pearson correlation coefficients being 0.64 and 0.68 for the matched and un-matched (mRNA 12h - protein 16h) time points, respectively. These results were essentially unchanged compared to LFQ quantification which yielded maximum correlation coefficients of 0.56 and 0.63 (mRNA 12h - protein 16h), respectively.

As expected, the difference between LFQ and iBAQ was even less pronounced when calculating local RNA-protein correlation: When considering IBAQ protein values, the pairwise Spearman correlations again showed a broad distribution over all mRNA/protein pairs, the mean correlation being 0.025 when no time-shift between mRNA and protein is assumed (LFQ-based mean: 0.009). Furthermore, the maximum mean mRNA/protein correlation achieved when assuming a time-lag of protein relative to mRNA is slightly smaller for iBAQ ($\rho=0.140$, 4h lag) when compared to LFQ ($\rho=0.158$, 6h lag). Finally, the fraction of significant positive correlations between mRNA/protein pairs for non-shifted and shifted time-courses is comparable (33.7% vs. 33.5%) when assessing correlation either based on LFQ or IBAQ values.

Taken together, iBAQ and LFQ quantification yielded very similar results for the RNA-protein correlation. Since a systematic comparison of iBAQ and LFQ values to our knowledge has not yet been carried out on a global transcriptome-proteome dataset, we included these results in the revised Supplemental material and briefly summarize them in the revised Results section. All modelling analyses were still performed using LFQ quantification, the main reason being that LFQ values, on average, had a smaller standard deviation between replicates (17 % of the mean) when compared to iBAQ values (24 % of the mean).

5) Were the RNAseq experiments performed using Spike-In controls? Burst in transcription at the end of MZD could lead to all transcripts being suddenly up regulated and normalization might produce artificial fold changes upon normalization. The authors should at least comment on this.

We did not use spike-in standard for the RNASeq analysis and assumed that all developmental time points had similar total RNA expression patterns, as was done in other dynamic transcriptome studies analyzing early *Drosophila* development (e.g., Daines et al., 2010; Graveley et al., 2011). We agree with the reviewer that spike-in normalization could have been beneficial to exclude artificial fold-changes in the mRNA dynamics. However, our RNA sequencing data argues against strong shifts in global gene expression, as we found that the total number of expressed transcripts with significant expression level does not appreciably change during MZT (Fig. S1A).

Furthermore, a recent study by Sandler & Stathopoulos used spike-in controls for absolute quantification of molecule numbers of 68 mRNAs in embryos. They found that large increases in zygotic mRNA expression due to genome activation during MZT are accompanied by similar decreases in maternal mRNA numbers due to induced degradation. This simultaneous up- and

down regulation indicated that the total number of transcripts per embryo may be quite stable.

To further substantiate this conclusion, we quantitatively compared RPKM values from our dataset (1,2,3 h, respectively) to corresponding absolute transcript counts from Sandler & Stathopoulos (NC10 [80min], NC13 late [120min], Gastrulation [180min], respectively). As shown in Figure 3, we found a strong correlation between both studies when considering mRNA numbers at each time point ($R=0.88$, 0.82 and 0.57 , respectively), as well as fold-changes between time points ($R=0.86$ and 0.92).

Figure 3 - Comparison of our RNA-Seq results (RPKM, x-axis) with absolute RNA quantifications reported in Sandler & Stathopoulos (y-axis) across 64 genes overlapping between both studies. The top and bottom rows show comparisons of corresponding time points and fold-changes between time points, respectively. The Pearson correlation coefficient (r) is indicated on the top left.

The conclusion of a stable expression across developmental stages is further supported by a recent single-embryo proteome survey (Hughes, ..., Krijgsveld, MSB 2014), in which protein expression around MZT (2-4h; i.e., before genome activation affects the proteome) was compared to a later time point (10-12h). The average fold-change over all proteins was close to zero, with strong increases and decreases in a few hundred zygotic and maternal proteins, respectively. This again suggests compensatory up- and downregulation during MZT and argues against strong shifts in global gene expression.

We mention these arguments in the revised Discussion and explicitly mention that we assumed the total mRNA and protein mass to be constant during development (also in response to comment 8 by reviewer # 2)

Minor points:

6) Some of the mass spec data were imputed, have the authors assessed the impact of the correlation between protein and mRNA with and without imputation?

We thank the reviewer for pointing out the need for clarification with respect to the usage of imputed protein values. Indeed, we only use imputed protein values while clustering mRNA/protein time-courses (Figure 1B of the manuscript). All following analysis, including assessment of mRNA/protein correlation as well as protein classification by ODE models, was performed without imputing protein values (i.e., the corresponding data points were removed from the dataset). We have now clarified the use of imputed protein values in the updated manuscript (section 5.8 and section 5.9 of the Material and Methods).

7) For the siRNA experiment, what is the rationale for choosing cut offs for RNAseq and proteome experiments? Is the cut off for proteomics data really based on “p value” and not adjusted for multiple testing? If this is case, it needs to be fixed.

Indeed the reviewer correctly points out that differentially expressed proteins are identified by uncorrected p-values. To our experience the currently applied multiple testing algorithms like Bonferroni or BH are easily applied, but too conservative for proteomics data. We would like to point the reviewer to a viewpoint published in *Proteomics* (Pascovici et al. 2016) that nicely illustrates the issues when performing proteomics data analysis. Indeed, while performing Benjamini-Hochberg correction on p-values almost all significantly changing proteins are lost (Figure 4).

Figure 4 - Impact of multiple testing correction on the identification of differentially expressed proteins in response to Hrb98DE knockdown. The histogram shows the frequency (y-axis) of p-values (x-axis) with (green) and without (blue) Benjamini-Hochberg

correction, respectively

As we do not correct for multiple testing in the proteomic data, we instead apply a more stringent cut-off to identify significantly changing proteins (p-value < 0.01, absolute fold-change > 1.5) when compared to the identification of significant changes on the mRNA level (BH corrected p-value < 0.05, absolute fold-change > 1.3). Please note also, that we also find differentially expressed protein without significant change in mRNA if an even detection criteria for significantly changing mRNA is considered. For example, if no fold-change cut-off is applied to mRNA data and the threshold for identifying significant changing mRNA is raised to BH corrected p-value < 0.1, we nevertheless identify 13 significantly changing protein without significant change on the level of mRNA.

We however notice the difficulty of comparing differentially regulated genes identified on transcriptome and proteome level due to discrepancies in the sample size, effect size and significance testing. We therefore check the validity of our results by visual inspection of mRNA and protein changes and indeed confirm the existence of genes with significant changes on protein level without changes on the level of mRNA (see Figure 7C of the manuscript)

8) While the manuscript is clearly written the Figures need to be improved. Some figures are cropped at the bottom (e.g. Fig.2B, Fig.4), some axis-label missing (Fig. 5).

We thank the reviewer for this valuable feedback, which we have used to improve figure quality.

Reviewer #2 (Remarks to the Author):

General remarks

The authors have investigated the mRNA-protein relationship during *Drosophila* embryogenesis using matching transcriptome and proteome data for 14 developmental time points. They observe that mRNA-protein correlations improve when considering delayed time points and that further improvements can be obtained using mass-action type models of translation and protein degradation. A notable discrepancy with a simple mass action model for a significant fraction of genes is due to the observed maternal-to-zygotic transition (MZT), after which the mass action models capture well the protein-mRNA relation. Finally, the authors have stated that proteins belonging to different kinetic models are enriched for different GO terms and their analysis has revealed RNA-binding protein Hrb98DE to be a post-transcriptional regulator of genes involved in sugar metabolism, which was validated by follow-up experiments.

1) Overall, this study did not delineate well its novelty.

Indeed, the biological conclusions of the authors about poor mRNA-protein correlations and kinetic models providing better insights about their relationship have been reported before (Jovanovic et al. 2015, Peshkin et al. 2015). Hence the most important biological message is already known. Furthermore, Peshkin et al. 2015 had generated and analysed a similar data for *Xenopus* and also reported differences for eth MZT transition. Hence the biological insights are confined to the fly community and to the RNA-binding protein Hrb98DE twist of the paper.

We thank the reviewer for his/her suggestion to better delineate the novelty of our work. This helped us to improve the presentation of our results. We agree that previous studies already used kinetic modeling of protein expression to gain better insights into the RNA-protein relationship. Compared to these previous studies, we go several steps further and describe a framework for the systematic discovery of post-transcriptional regulation mechanisms controlling a biological process of interest. Specifically, we quantify our model fitting results, derive specific lists of proteins predicted to be under post-transcriptional control and make predictions about mechanisms fine-tuning their protein levels. The main novelties of our approach are highlighted in bold below.

Our framework involves several steps, including state-of-the-art model rejection analyses using χ^2 and Durbin-Watson tests, by which we identify feasible model variants for each protein. In response to the reviewers' comments, we now refined these methods in the revision by testing for normality in the residuals (comment 20), using multiple-testing correction by applying a parametric bootstrap strategy (comment 21). During the revision, we now confirm the performance of such model rejection analysis using in silico benchmarks (see comment #2). In contrast, previous studies mainly relied on qualitative metrics such as the cosine distance (Peshkin et al.). Unlike earlier work, we proceed using a likelihood ratio test for model selection to identify the most appropriate model variant for each protein, thereby unequivocally assigning each protein to a certain class of regulation. The resulting lists of proteins were then subjected to gene ontology and RBP sequence motif analyses in order to: (i) discover biological processes under post-transcriptional control and (ii) identify putative post-transcriptional regulators controlling large sets of target genes. We confirm the predictive power of our framework by showing that a Hrb98DE knockdown leads to deregulation at the level of protein expression and splicing, with enriched effects at genes functioning in glucose metabolism.

In our opinion, the biological insights are thus not simply confined to the fly community and to the RNA-binding protein Hrb98DE twist of the paper. Instead, our work describes a thorough quantitative analysis framework, that represents a blueprint for the discovery of novel post-transcriptional regulation mechanisms. In response to comment #2 below, we show that this framework is robust against experimental variation.

2) Methodologically, the modeling approach has not been compared against alternative existing approaches (eg Teo et al 2014, Peshkin et al. 2015) nor it has been thoroughly benchmarked against simulations and independent data. Also, the model is based on strong modelling assumptions (absolute vs relative concentrations), and non-assessed assumptions important for statistical testing (normality). They were also errors in the python code. Therefore, the modelling itself will not be likely reused.

Altogether, I recommend rejection of the paper. It would need re-working of the model, a clear delineation of the contribution of this study to state-of-the-art rather than a mere referencing (additional important references are Robles et al PLoS Genet, 2014, Cheng et al Mol Syst Biol 2016 and Lau et al., Nature comms, 2018.), and the retargeting to a more specialized journal. I hope my detailed comments below will help the authors to this end.

We agree with the reviewers' criticism, and therefore decided to better put our work in context, test it against independent benchmark data and checked several model assumptions. The results of these analyses are briefly summarized below:

We included an *in silico* benchmark, in which we simulated hundreds of time courses for transcriptionally and post-transcriptionally regulated proteins using randomly sampled parameter values. We add noise to this *in silico* data and then use our model fitting framework to identify post-transcriptionally regulated proteins (Supplemental Text S3). At all tested noise levels, we find an excellent precision (> 0.9) of our model-based classification, implying that a protein will most likely be regulated at the post-transcriptional level if all models have to be rejected. The recall was lower, but still significantly higher than picking proteins by chance, implying that we did not identify all proteins for which post-transcriptional regulation was assumed in the simulation. Further analyses suggest that, in many cases, proteins with simulated post-transcriptional regulation can still be fitted by adjusting parameters in the production model, i.e., post-transcriptional regulation cannot be unequivocally identified from the time courses. If we correct for such non-identifiability, we obtain an excellent recall (> 0.99) at all noise levels. Taken together, our analysis suggests that our framework reliably identified and rejected post-transcriptional regulation whenever this kind of regulation leaves a unique footprint in a protein time course.

Given these high precision and recall values, we felt it is no absolute requirement to compare our method against existing approaches and that such analysis is beyond the scope of this work. This decision was further influenced by technical difficulties: The publically available implementation of the time-discrete model to identify RNA- or protein-level regulation (PECA) was not functional, even if provided example input files were used. Unfortunately the developers of the package did not respond to our inquiry about possible causes for this non-functionality. To address the reviewers' comment, we more clearly delineated the novelty of our work (see response to comment above), and included the suggested studies (Teo, Lau, Robbles, Cheng) in our Discussion.

The reviewers' concerns about modelling assumptions are addressed in response to more detail comments below: For a discussion of the issue of absolute compared to relative concentrations see comment #8. We further assessed normality of residuals (comment #20) and checked for errors in the python code (comments #13-#17). The latter responses demonstrate that our framework can very well be reused.

Minor points

3. Abstract. "the direct correlation between mRNA levels and protein abundances is moderate in most studies" Most studies are comparing steady-state conditions. For steady-state data, using ODE would not help. Hence, the argument does not help for "most studies".

We agree with the reviewer that for steady-state data indeed ODE models would not be of

benefit. This is why we explicitly stated in the last sentence of the Abstract that our framework is suitable “for the identification of post-transcriptional gene regulation for large-scale time-resolved mRNA and protein expression patterns”. Thus, we did not claim that our approach is helpful for the analysis of steady state data.

To further avoid confusion at the begin of the Abstract, we now formulated the second sentence more precisely as: “We asked whether the relation of mRNA and protein *during a dynamic developmental transition* can be better explained by simple mathematical models based on ordinary differential equations (ODEs) incorporating a temporal dimension.” We did not change the first sentence, as our statement of a moderate mRNA-protein correlation in most studies remains valid.

4. Abstract. “ODE models describing protein translation and degradation fit 81% of mRNA-protein profiles” is unclear. One needs to read the paper to understand it. I understand that no correction for multiple testing was done (see below).

We agree with the reviewer that the expression is open to misunderstanding and simplified the terminology accordingly.

5. The discussion of the correlation of present RNA-seq data with previously published one is a bit lengthy and off the core topic of this paper. It could be made more succinct.

We agree and shortened the corresponding paragraphs accordingly.

6. From the publication and website, MISO seems to only estimate isoform distribution of genes with multiple isoforms (PSI). How did they get values for the whole locus?

Indeed MISO estimates differential splicing of individual exons. However, most genes contain multiple exons. In order to determine differential splicing for the complete locus we required at least one of the exons to significantly change upon knock down. We specifically mention this in the revised version of our manuscript.

7. The iBAQ quantification has been developed for the purpose of absolute quantification. The authors should try with iBAQ instead of LFQ and report whether it improves the results.

According to comments raised by Reviewer #1 as well as Reviewer #2, we have now checked the impact of using IBAQ quantification on the correlation of mRNA and protein. Please refer to our response to comments raised by Reviewer #1 for a more detailed discussion of this issue.

8. Since the protein and RNA quantifications are relative to each other, the ODEs assume that the total RNA mass and the total protein mass does not change over time. Is that realistic? That assumption should be made explicit and discussed if not assessed. Also RNA-seq protocol depletes for ribosomal RNAs.

This reviewer comment is related to major point 5 of reviewer #1 who suggested spike-in controls for absolute quantification. In fact, we did not express mRNA and protein in absolute molecule numbers, but in relative units as was done in previous related work (e.g., Daines et al., 2010; Graveley et al., 2011). Due to our normalization of mRNA and protein counts by the total count at that time point it is possible that artificial fold-changes arise from total mRNA or protein mass changes.

As discussed in our response to reviewer #1, we consider strong mass changes as unlikely: At the mRNA level, we compared our RNASeq time courses to a recent medium-scale study in which spike-in standards were used for absolute quantification and found a strong correlation. At the protein level, a proteomics study compared expression around and after MZT at the single-embryo level. The average fold-change over all proteins was close to zero, with strong increases and decreases in a few hundred zygotic and maternal proteins, respectively. Collectively, these observations argue against strong shifts in global gene expression during embryonic development.

We explicitly discuss our assumption in the revised Discussion and mention that the depletion of ribosomal RNAs might mask total RNA mass changes.

9. Why were the SR2+ data analysed differently (read mapping and counting, no isoform distribution estimation)?

The reviewer is correct that differences exist in the processing of the RNA-Seq data from full embryos compared to S2R+ cells. This difference came as a result of minor modifications in our data processing pipeline, in which the 'featurecounts' method was replaced by the updated 'htseq-count' method. Both methods however use equivalent quantification procedures of RNA read counts, and have been applied using the same parameters. These minor differences in the RNA-Seq processing pipeline therefore do not impact the overall result of our study. We clarify this in the revised manuscript.

Concerning the isoform distribution, it is important to keep in mind that the S2R+ and *in vivo* samples were analyzed using different sequencing protocols (different read length). A longer read length while sequencing allows for more reliable detection of splicing changes and was used in the SR2+ Hrb98DE knockdown experiment (75bp), as this factor is known to control alternative splicing at the post-transcriptional level. On the contrary, shorter read length allows for higher throughput and was therefore used for the embryo dataset due to large number of

time points and replicates, at the expense of a comprehensive characterization of alternative splicing. We clarify this in the revised manuscript (Section 5.7).

10. “In contrast to the Pearson correlation coefficient, Spearman correlation makes no assumptions about normality of the data or linearity in the dependence of the variables (de Sousa Abreu et al., 2009; Tobias Maier et al., 2009)”. The ref (de Sousa Abreu et al., 2009; Tobias Maier et al., 2009) are not appropriate for this statement. It is not their work but Spearman’s.

We thank the reviewer for pointing out this inaccuracy. Spearman is a very well established measure of correlation and therefore decided to remove the previously mentioned citations.

11. “We included only significantly changing proteins (n = 866) to avoid that random variations around the mean might mask existing trends between mRNA and protein”. What is the statistical test and multiple testing procedure applied? Which significance cutoff was used? The method refers back to Casas-Vila et al. Gen Res 2017. However, i) these informations are important to understand this paper so they must be recapitulated here and ii) Casas-Vila have multiple ways of selecting differentially regulated proteins. I could not find one that had the same number (866) than here.

Casas-Vila et al. 2017 found 1644 developmentally regulated protein groups in embryogenesis, applying the statistical test described in the supplemental material within (Casas-Vila et al. 2017). Since in our study we only consider proteins with a maximum of 4 missing measurements across the full time-course (NaN values), the overlap between these two sets resulted in 866 significantly changing proteins. In order to avoid any further confusion about statistical testing, we have now adapted Figure 2A to include only proteins which show high absolute fold changes (top 500) instead of using significantly changing proteins. This further allows for a better comparison between LFQ and iBAQ quantified proteins. Results obtained by this strategy do not impact the overall conclusions obtained from the previous approach based on statistical testing.

12. It is not the “heavy-side” function, but the “Heaviside” function, named after its inventor.

We thank the reviewer for noticing this error, which we have now corrected in the manuscript.

13. Equation (3) must be $\exp(-\lambda t)$ and not $\exp(\lambda t)$.

Indeed, Equation (3) in the manuscript was malformed and is now corrected. We appreciate

the detail, with which the reviewer inspected our manuscript and have made the corresponding corrections. Also in response to multiple following comments, we additionally provided a more extensive and point-by-point derivation of model equations, with the aim to more clearly communicate our modelling approach (Supplemental Text S2).

14. Equation 4: $h(-t\tau)$ should be $h(t\tau)$?

As the value of t in Equation (4) is equal to zero, indeed both expressions are correct. We have now adapted the description of our modelling approach accordingly to clarify for this potential misunderstanding. As mentioned before we also provide a point-by-point derivation of model equations in Supplemental Text S2.

15. If Equation (4) is correct (I think it is up to point #12) then the python code line 29 and line 94, which calculates “c” in solve_modelD and in solve_modelC in the file solve_model.py are wrong. This must be corrected and the analysis re-done.

Again we want to thank the reviewer for the detailed inspection of our manuscript together with the supplied source code. Based on these comments, we have once again thoroughly checked all equations and their implementation. Indeed, we discovered that Equation (4) provided in the manuscript was badly formatted, while in the Python code the variable “c” was calculated correctly. As mentioned before, we now provide a more detailed derivation of model equations in Supplemental Text S2.

16. The code and Suppl Table 1 should be provided (I got them afterward)

We agree this critical information needs to be provided. Accordingly, we included it in the revised manuscript.

17. According to the code, $u(t) = mt + b$ is assumed in each interval. Hence interval indices should be introduced for the slope and intercept for clarity.

We thank the reviewer for pointing out that introducing interval indices may increase the clarity and reproducibility of model equations. Accordingly we have taken this into account while providing a more detailed derivation of model equations in Supplemental Text S2.

18. How good does this model fit overall to the data? A global scatterplot must be provided.

According to the reviewers' comment we now provide global scatter plots of modeled vs measured protein expression in Supplemental Figure S3. Analyzing all four proposed models and the genes best described by them, we find that the Pearson correlation between modeled and measured protein expression is above 0.99. For mRNA/protein pairs for which all four proposed models needed to be rejected, we made use of simulated protein

expression resulting from the best-fits of the 'delay'-model (which contains most degrees of freedom) and compared these values to the data. For proteins falling into the 'rejected' category, the Pearson correlation drops to a value of 0.92, demonstrating that even the most complex model fails to fully describe these data.

19. The model is written for protein and RNA expression in the natural scale. However, both for RNA and protein, noise is typically multiplicative in the natural scale. Therefore, the fitting can suffer from heteroscedasticity, whereby high expression data points, which also present largest noise, have strongest influence on the fits. Is that the reasons why weighted least squares are used? If so, how the weights are computed should be made clear.

Heteroscedasticity and the multiplicative nature of the noise can be accounted for by a logarithmic data transformation or by assuming relative errors (which are proportional to the mean signal). We used a linear model with a relative (but without absolute) error to describe the relationship between noise levels and expression values. The estimated slope (0.17) corresponds to 17% relative error which is used for weighted least squares. A similar error model was also used when fitting other models describing RNA and protein dynamics, e.g., in the benchmark models for gene regulatory networks in the so-called DREAM challenges (Steiert et al. 2012, Meyer et al. 2014). We appreciate the comment made by reviewer #2 and clarified the description of this part of analysis in the new version of our paper.

20. Normality assumption of the residuals is important for the chi-squared test. A quantile-quantile plot of the residual must be provided. If Normality is not met, alternative non-parametric testing strategies must be investigated and applied (eg Bootstrap of cases or of residuals). The same issue applies to the Durbin-Watson test.

We agree that the assignment of mRNA-protein pairs to regulatory mechanisms may be problematic if normality is not met. For instance, non-normality (e.g. caused by outliers in the data) might lead to rejecting basic mechanisms by the chi-squared test or by the Durbin-Watson test. If all basic mechanisms would be rejected, we would consequently incorrectly claim post-transcriptional regulation. To address the reviewer comment, we analyzed for normality as explained in the following.

In 3007 out of 3761 mRNA/pairs the most complex delay model could not be rejected. As all our models are nested (stationary < degradation < production < delay), we expect only a small number of models being rejected without simultaneous rejection of the 'full' delay model. Indeed for only 52 mRNA-protein pairs, the delay model needed to be rejected while other, more simple models remained possible. This implies that the set of post-transcriptionally regulated proteins identified by model rejection analysis via the combined chi-squared and Durbin-Watson test is to a large extent dependent only on the rejection of the delay model.

To simplify the analysis, we therefore focused on analysis on the 3007 proteins explained by the delay model in order to assess whether normality assumptions of the residuals are fulfilled. We initially analyzed each time course fit individually and used the Shapiro-Wilk test to assess for normality of residuals. Based on this criterion, in only 6.2% of all 3007 mRNA/protein pairs non-normality of the residuals was observed with a p-value of 0.05. As a second analysis, we more globally analyzed the residuals across all time points and proteins together and plotted them as a histogram which again indicated good agreement with a normal distribution (Fig. 5, left). This conclusion is further supported by an almost linear quantile-quantile plot which again indicates a close match of the distribution to the standard normal distribution (Fig. 5, right).

Figure 5 - Histogram and Q-Q-Plot of residuals over all time points and all 3007 proteins fitted by the delay model indicate a close-to-normal distribution.

We therefore conclude that for almost all mRNA/protein pairs the assumption of normality of residuals holds. Non-normality in specific cases should not impact the overall results produced by our model rejection approach. However, in order to prevent claiming post-transcriptional regulation erroneously only due to outliers/non-normality, in the new version of our manuscript, we applied more stringent model rejection criteria, in which nominal p-values for chi-squared and Durbin-Watson test were further corrected for multiple testing (see next paragraph).

21. The cut-off on type I error was based on nominal p-value ($\alpha = 0.05$). The authors should consider multiple testing corrections.

According to the reviewers we have now considered multiple testing corrections for model rejection analysis. Multiple testing is usually applied to reduce the absolute number of rejections (false positives), based on corrected p-values. P-values in the case of the chi-

squared test are readily available via the respective subroutine (`scipy.stats.chi2`), while this was not the case for the Durbin-Watson test. In order to be able to correct for multiple testing also for the Durbin-Watson test, we therefore implemented a parametric bootstrap procedure ($n=1000$) from which p-values can be calculated based on the empirical distribution. For both test criteria, obtained by the parametric bootstrap approach, p-values were then corrected using Benjamini-Hochberg correction.

For a more stringent significance threshold - e.g. when applying multiple testing correction - one would expect a lesser number of proteins for which all model variants have to be rejected. In accordance with this expectation, after multiple testing correction less mRNA/protein pairs are classified as post-transcriptionally regulated after applying multiple-testing criteria (16% pt-regulated, as opposed to 19% in the previous analysis). We therefore believe that the updated list contains putative post-transcriptionally regulated proteins of higher confidence when compared to the previous list. The overall distribution of the remaining not-rejected mRNA/protein pairs across the different model classes remains similar.

The revised manuscript was updated according to these refined classification results. Subsequent analysis steps (e.g., GO term and motif enrichment analysis) which were based on the classification of mRNA/protein pairs have been updated as well, without any major changes in the conclusions drawn.

22. The initial value y_0 should be fitted rather than forced to be equal to the first measurement value. The reason is that every measured value entails noise. Errors in y_0 propagate to all data points. They may also be a strong driver of the anti-correlation seen in Fig 4B.

As stated in the results section of our original Manuscript, the value of y_0 is estimated together with other model parameters instead of being equal to the first measurement value. We thank the reviewer for pointing out this potential miscommunication and have now provided a more comprehensive summary of model formulations together with details on parameter estimation (Supplementary Text S2).

23. Fig 4B should be backed by independent measures. For instance, by comparing half-lives from other measurements of protein half-life and scatter plotting. Also, the anti-correlation might be a statistical artefact (correlation of estimation errors). One way to address this issue is to check that the trend holds when production rates and half-lives are estimated from independent subsets of the data (e.g. using the odd data points for production and the even data points for half-life).

We carried out the proposed cross-validation, and found that the correlation in Fig. 4B is no longer visible if production and degradation are estimated independently. Hence we agree with the reviewer that this may be an artifact of parameter correlations in the model and

have therefore removed the corresponding paragraph from our manuscript, as this was not a major result of our work. Accordingly, the comparison to independent half-life measurements was also obsolete.

Reviewer #3 (Remarks to the Author):

This paper is a treasure trove of new information that should stimulate much productive further investigation. The most important point - for this reviewer - is the demonstration that the poor correlation between transcript and protein titers can, in a majority of cases, be explained without resort to idiosyncratic (i.e. gene-specific) post-transcriptional regulation. The authors show that relatively simple models containing delays and accounting for protein degradation do the trick. The robustness of the allocation of various genes to these separate models might have been strengthened by additional Monte Carlo simulations, but I consider that a quibble; the particular assignments aren't the great take-away lesson. The success in reconciling the majority of the cases is -- and that result should stimulate much new work by the authors and by others.

The HRB98DE results I count as an additional bonus. They are persuasive and, again, I think they will stimulate others to try to do similar things.

The manuscript is well written. A few figure legends could have been clearer, but they can be worked out.

We thank the reviewer for supporting our work and for his/her constructive criticism. In order to address the comments, we reworked the figure legends and tested our model selection using an in silico benchmark, in which assumed different types of (post-)transcriptional regulation and randomly sampled model parameters as suggested (see new Supplemental Text S3). In addition, we improve the identification of post-transcriptional regulation using a parametric bootstrap approach and adjust by multiple testing procedures.

References

- Blanchette, M., Green, R. E., MacArthur, S., Brooks, A. N., Brenner, S. E., Eisen, M. B., & Rio, D. C. (2009). Genome-wide Analysis of Alternative Pre-mRNA Splicing and RNA-Binding Specificities of the *Drosophila* hnRNP A/B Family Members. *Molecular Cell*, 33(4), 438–449. <http://doi.org/10.1016/j.molcel.2009.01.022>
- Brooks, A. N., Duff, M. O., May, G., Yang, L., Bolisetty, M., Landolin, J., ... Brenner, S. E. (2015). Regulation of alternative splicing in *Drosophila* by 56 RNA binding proteins. *Genome Research*, 25(11), 1771–1780. <http://doi.org/10.1101/gr.192518.115>
- Bushati, N., Stark, A., Brennecke, J., & Cohen, S. M. (2008). Temporal Reciprocity of miRNAs and Their Targets during the Maternal-to-Zygotic Transition in *Drosophila*. *Current Biology*, 18(7), 501–506. <http://doi.org/10.1016/j.cub.2008.02.081>
- Jeremy E. Sandler and Angelike Stathopoulos. Quantitative Single-Embryo Profile of *Drosophila* Genome Activation and the Dorsal–Ventral Patterning Network
- Meyer, P., Cokelaer, T., Chandran, D., Kim, K. H., Loh, P.-R., Tucker, G., ... Saez-Rodriguez, J. (2014). Network topology and parameter estimation: from experimental design methods to gene regulatory network kinetics using a community based approach. *BMC Systems Biology*, 8(1), 13. <http://doi.org/10.1186/1752-0509-8-13>
- Pascovici, D., Handler, D. C. L., Wu, J. X., & Haynes, P. A. (2016). Multiple testing corrections in quantitative proteomics: A useful but blunt tool. *Proteomics*, 16(18), 2448–2453. <http://doi.org/10.1002/pmic.201600044>
- Steiert, B., Raue, A., Timmer, J., & Kreutz, C. (2012). Experimental design for parameter estimation of gene regulatory networks. *PLoS ONE*, 7(7). <http://doi.org/10.1371/journal.pone.0040052>
- Ji, Y., & Tulin, A. V. (2016). Poly(ADP-Ribosyl)ation of hnRNP A1 Protein Controls Translational Repression in *Drosophila*. *Molecular and Cellular Biology*, 36(19), 2476–2486. <http://doi.org/10.1128/MCB.00207-16>
- Valencia-Sanchez, M. A., Liu, J., Hannon, G. J., & Parker, R. (2006). Control of translation and mRNA degradation by miRNAs and siRNAs. *Genes and Development*. <http://doi.org/10.1101/gad.1399806>

REVIEWERS' COMMENTS:

Reviewer #1 (Remarks to the Author):

I appreciate the efforts made by the author to generally address my comments in a definitely improved version of the manuscript. I also appreciate the efforts to validate in vivo the role of HRB98D during embryo development and I understand the technical difficulties that prevented the authors to perform a deeper analysis. However, my concerns remain regarding their claim of having validated the role of HRB98D in post-transcriptional regulation of sugar metabolism in early embryogenesis. Their claim is currently based on two pieces of evidence:

- (i) the overlap between proteins affected by HRB98D knockdown in vivo and the proteins that they define as post-transcriptionally regulated during embryogenesis ('rejected'). They show this overlap in Figure 7A and they express it in term of '% of differentially expressed' proteins. If I read the graph correctly, this means that a bit more than 5% of the proteins affected by HRB98D knockdown were classified as 'rejected', and thus post-transcriptionally regulated, by their models. This equals to 5% of the 75 proteins that they found to be differentially expressed upon HRB98D knockdown, which means 4 proteins!! If my interpretation is correct, the authors must indicate that although significant by hypergeometric test, their statement is based only on 4 proteins. This should be indicated in the main text as well as in Figure 7A. Additionally, for transparency, the authors should test the overlap also with other categories of proteins ('stationary', 'degradation', 'production', etc...) and display it as additional bars in the same plot. This of course should also be applied to Figure S6C.

- (ii) the knockdown experiment performed in SR2+ cells, where they achieved much higher knockdown efficiency. Despite the higher knockdown efficiency, they found again very few proteins affected in their abundance (only 40), which is a bit surprising for a supposed post-transcriptional regulator of protein abundance. Surprisingly, in this case they did not check the overlap with the 'rejected' proteins. The authors should test this and report the result even if negative. Finally, because of a substantial lack of signal at the protein level, the authors resort in checking for alternative splicing events and indeed found among the affected genes some involved in glucose metabolism, and some that overlap (13 out of 40) with significantly affected proteins. Although this is reassuring to see, I do not appreciate the novelty of the finding since the mammalian homologue of HRB98D is a well known splicing factor, see for example:
<https://www.ncbi.nlm.nih.gov/pubmed/1531115>
<https://www.ncbi.nlm.nih.gov/pubmed/23863836>

In summary, I believe that the validation experiments are still very weak and the authors should at minimum include the changes that I suggested above to improve transparency, and drastically down tone their statements regarding validation both in the abstract and at the end of the result paragraph where they state:

In summary, the identification of post-transcriptionally regulated genes together with the subsequent bioinformatic analysis suggested post-transcriptional regulation of glucose metabolism by the RBP Hrb98DE, which we were able to confirm experimentally.

There are also a couple of imprecisions and mistakes that should be corrected throughout the manuscript (here two examples, but there might be more):

- Figure 4, the legend states 'Left' mRpL23 while the plot show su(Hw);
- Page 14, This agrees with previous evidence showing post-transcriptional control of

genes functioning in glucose metabolism (Robles et al., 2014). It is not clear in which context, please specify.

Reviewer #2 (Remarks to the Author):

The manuscript has improved. The model is now much better explained and statistically justified. All my points have been addressed with satisfaction.

Regarding the impact, the second sentence of the abstract remains very misleading: "We asked whether the relation of mRNA and protein during a dynamic developmental transition can be better explained by simple mathematical models based on ordinary differential equations (ODEs) incorporating a temporal dimension.", which is a fundamental question – but already answered. As the authors replied to my point #1, the actual novelty is the follow-up step to detect known post-transcriptional regulatory elements enriched in some gene class (using some standard tool). The authors argue that ODE modeling is done now with greater care than previous papers, but these are no significant conceptual advances.

The abstract and introduction should be articulated around the actual novelty. Some better version of "The role of post-transcriptional regulation play in development remains poorly understood. To detect such regulation, we propose to detect genes with apparent strong post-transcriptional regulation by jointly modeling RNA and protein time series and search these genes for enriched RBP motifs. Applying this approach to D. mel. we discovered...". A paper title along these lines would also be helpful.

Moreover, I also think that to avoid confusions, the delineation with the state of the art, as motivation for the present work, belongs to the introduction (comparison with Teo, Lau, Robbles, Cheng) and not to discussion.

It will then be acceptable. I leave the appreciation of the relevance for the nature communication readership to the editors.

Response to reviewers comments

Reviewer #1 (Remarks to the Author):

I appreciate the efforts made by the author to generally address my comments in a definitely improved version of the manuscript. I also appreciate the efforts to validate in vivo the role of HRB98D during embryo development and I understand the technical difficulties that prevented the authors to perform a deeper analysis. However, my concerns remain regarding their claim of having validated the role of HRB98D in post-transcriptional regulation of sugar metabolism in early embryogenesis. Their claim is currently based on two pieces of evidence:

- (i) the overlap between proteins affected by HRB98D knockdown in vivo and the proteins that they define as post-transcriptionally regulated during embryogenesis ('rejected'). They show this overlap in Figure 7A and they express it in term of '% of differentially expressed' proteins. If I read the graph correctly, this means that a bit more than 5% of the proteins affected by HRB98D knockdown were classified as 'rejected', and thus post-transcriptionally regulated, by their models. This equals to 5% of the 75 proteins that they found to be differentially expressed upon HRB98D knockdown, which means 4 proteins!! If my interpretation is correct, the authors must indicate that although significant by hypergeometric test, their statement is based only on 4 proteins. This should be indicated in the main text as well as in Figure 7A. Additionally, for transparency, the authors should test the overlap also with other categories of proteins ('stationary', 'degradation', 'production', etc...) and display it as additional bars in the same plot. This of course should also be applied to Figure S6C.

We apologize that our previous presentation of these results might have been misleading: Figure 7A (and Figure S6C) shows the percentage of differentially expressed genes within the the set of post-transcriptionally regulated genes (compared to background genes as well as the set of Hrb98DE targets).

In the case of KD2, the figure therefore indicates that 5.3% of 456 putative post-transcriptionally regulated genes are differentially expressed, compared to only 3.0% of differentially expressed genes in the set of background genes. In explicit numbers, this relates to 24 observed differentially expressed genes within the set of post-transcriptionally regulated genes compared to a number of 13.26 expected genes. Hence, the significant p-value of 1.6e-3 (hypergeometric test). We have now modified these figures to avoid confusion and have implemented the suggestion from the reviewer to include all models in Figure 7A (and Figure S6C).

- (ii) the knockdown experiment performed in SR2+ cells, where they achieved much higher knockdown efficiency. Despite the higher knockdown efficiency, they found again very few proteins

affected in their abundance (only 40), which is a bit surprising for a supposed post-transcriptional regulator of protein abundance. Surprisingly, in this case they did not check the overlap with the 'rejected' proteins. The authors should test this and report the result even if negative. Finally, because of a substantial lack of signal at the protein level, the authors resort in checking for alternative splicing events and indeed found among the affected genes some involved in glucose metabolism, and some that overlap (13 out of 40) with significantly affected proteins. Although this is reassuring to see, I do not appreciate the novelty of the finding since the mammalian homologue of HRB98D is a well known splicing factor, see for example: <https://www.ncbi.nlm.nih.gov/pubmed/1531115>
<https://www.ncbi.nlm.nih.gov/pubmed/23863836>

We performed now the statistical tests for overlap between differentially changing proteins and differentially spliced genes to the group of rejected proteins. We do not find any overlap and mention this result in the manuscript.

In summary, I believe that the validation experiments are still very weak and the authors should at minimum include the changes that I suggested above to improve transparency, and drastically down tone their statements regarding validation both in the abstract and at the end of the result paragraph where they state:

In summary, the identification of post-transcriptionally regulated genes together with the subsequent bioinformatic analysis suggested post-transcriptional regulation of glucose metabolism by the RBP Hrb98DE, which we were able to confirm experimentally.

As suggested by the reviewer we rephrased parts of the manuscript to eliminate overstatements.

There are also a couple of imprecisions and mistakes that should be corrected throughout the manuscript (here two examples, but there might be more):

- Figure 4, the legend states 'Left' mRpL23 while the plot show su(Hw);

We thank the reviewer for this information and have corrected the error.

- Page 14, This agrees with previous evidence showing post-transcriptional control of genes functioning in glucose metabolism (Robles et al., 2014). It is not clear in which context, please specify.

We removed this statement in the process of shortening the MS.

Reviewer #2 (Remarks to the Author):

The manuscript has improved. The model is now much better explained and statistically justified. All my points have been addressed with satisfaction.

We thank the reviewer for the previous helpful comments and agree that they did improve the manuscript.

Regarding the impact, the second sentence of the abstract remains very misleading: “We asked whether the relation of mRNA and protein during a dynamic developmental transition can be better explained by simple mathematical models based on ordinary differential equations (ODEs) incorporating a temporal dimension.”, which is a fundamental question – but already answered. As the authors replied to my point #1, the actual novelty is the follow-up step to detect known post-transcriptional regulatory elements enriched in some gene class (using some standard tool). The authors argue that ODE modeling is done now with greater care than previous papers, but these are no significant conceptual advances.

To avoid this putative overstatement, we have reformulated the beginning of the abstract and dropped this sentence completely.

The abstract and introduction should be articulated around the actual novelty. Some better version of “The role of post-transcriptional regulation play in development remains poorly understood. To detect such regulation, we propose to detect genes with apparent strong post-transcriptional regulation by jointly modeling RNA and protein time series and search these genes for enriched RBP motifs. Applying this approach to *D. mel.* we discovered...”. A paper title along these lines would also be helpful.

We thank the reviewer for the suggestion and modified our abstract along these lines. We furthermore accepted the title changes provided by the editor.

Moreover, I also think that to avoid confusions, the delineation with the state of the art, as motivation for the present work, belongs to the introduction (**comparison with Teo, Lau, Robbles, Cheng**) and not to discussion.

As suggested by the reviewer, we moved this part to the Introduction.

It will then be acceptable. I leave the appreciation of the relevance for the nature communication readership to the editors.